# Harnessing PROTAC technology to combat stress hormone receptor activation

Mahshid Gazorpak [1,2,11], Karina M. Hugentobler [3,11], Dominique Paul[4], Pierre-Luc Germain [1,4,5], Miriam Kretschmer [1,2], Iryna Ivanova [1], Selina Frei[1], Kei Mathis [1], Remo Rudolf [1], Sergio Mompart Barrenechea [1], Vincent Fischer [1,2], Xiaohan Xue [6], Aleksandra L. Ptaszek [7], Julian Holzinger [7], Mattia Privitera [8], Andreas Hierlemann [6], Onno C. Meijer [9], Robert Konrat[10], Erick M. Carreira[3], Johannes Bohacek [2,8] & Katharina Gapp [1,2] ✉

Counteracting the overactivation of glucocorticoid receptors (GR) is an important therapeutic goal in stress-related psychiatry and beyond. The only clinically approved GR antagonist lacks selectivity and induces unwanted side effects. To complement existing tools of small-molecule-based inhibitors, we present a highly potent, catalytically-driven GR degrader, KH-103, based on proteolysis-targeting chimera technology. This selective degrader enables immediate and reversible GR depletion that is independent of genetic manipulation and circumvents transcriptional adaptations to inhibition. KH-103 achieves passive inhibition, preventing agonistic induction of gene expression, and significantly averts the GR's genomic effects compared to two currently available inhibitors. Application in primary-neuron cultures revealed the dependency of a glucocorticoid-induced increase in spontaneous calcium activity on GR. Finally, we present a proof of concept for application in vivo. KH-103 opens opportunities for a more lucid interpretation of GR functions with translational potential.

Excessive signalling by glucocorticoid (GC) hormones is linked to many pathologies[1]. Exposure to stressful situations triggers activation of the hypothalamus-pituitary-adrenal (HPA) axis, which results in the release of GCs into the bloodstream. Under healthy conditions, GCs support adaptation through modulation of all major organ systems,

including the brain, in large measure via the glucocorticoid receptor (GR)[2,3]. GRs act as a transcription factor (TF): upon activation, cytosolic GR translocates to the nucleus and binds directly or indirectly to genomic response elements as a homo or heterodimer[4]. This process can result in the promotion or inhibition of transcription and/or alter

[1]Laboratory of Epigenetics and Neuroendocrinology, Institute for Neuroscience, Department of Health Science and Technology, ETH Zürich, 8057 Zürich, Switzerland. [2]Neuroscience Center Zürich, ETH Zürich and University of Zürich, 8057 Zürich, Switzerland. [3]Laboratory of Organic Chemistry, Department of Chemistry and Applied Biosciences, ETH Zürich, 8093 Zürich, Switzerland. [4]Lab of Statistical Bioinformatics, University of Zürich, 8057 Zürich, Switzerland. [5]Computational Neurogenomics, Institute for Neuroscience, Department of Health Science and Technology, ETH Zürich, 8057 Zürich, Switzerland. [6]Bio Engineering Laboratory, Department of Biosystems Science and Engineering, ETH Zürich, 4056 Basel, Switzerland. [7]Christian Doppler Laboratory for High-Content Structural Biology and Biotechnology, Max Perutz Laboratories, Department of Structural and Computational Biology, University of Vienna, Campus Vienna Biocenter 5, 1030 Vienna, Austria. [8]Laboratory of Molecular and Behavioral Neuroscience, Institute for Neuroscience, Department of Health Science and Technology, ETH Zürich, 8057 Zürich, Switzerland. [9]Department of Medicine, Division of Endocrinology, Leiden University Medical Center, 2300 RA Leiden, the Netherlands. [10]Department of Structural and Computational Biology, University of Vienna, Campus Vienna Biocenter 5, 1030 Vienna, Austria. [11]These authors contributed equally: Mahshid Gazorpak, Karina M. Hugentobler. ✉e-mail: katharina.gapp@hest.ethz.ch

chromatin accessibility to prime gene transcription[5]. Whether this binding then leads to enhancer activation and nearby gene transcription depends on complex interactions with cofactors determined by the local motif composition[6].

Dysregulation of GR levels and GC secretion are hallmarks of chronic stress-induced conditions, including depression, but also neurodegenerative diseases[7–12]. For example, in patients with post-traumatic stress disorder, hypersensitive GR and associated reductions in GC levels have been repeatedly found[7]. Besides implications in neuropsychiatric diseases, altered GC levels play a major role in the progression and metastasis of various types of cancer, Cushing syndrome, and metabolic disease[13].

Treatment strategies have already been explored that attenuate GR signalling. For instance, in patients with psychotic depression, treatment with the clinically available but non-selective GR antagonist mifepristone (MIF) showed promising results, such as rapidly improving depression symptoms[14,15]. However, binding of MIF to the progesterone and androgen receptor and long-term high dosage requirements are considerable disadvantages[13]. In rodents, administration of MIF has proven beneficial to mitigate stress pathology and has been used to study the implication of GR in the dysregulation of the stress response[14,16]. Despite such effects, the outcome of MIF is difficult to interpret, as the MIF-GR complex recruits transcriptional coregulators and may show (partial) agonism[17,18]. An additional complication is its highly different plasma half-life between rodents and humans[19,20].

To address the need to better target aberrant GR activation in various disorders, recent research has explored GR antagonists and mixed antagonists/agonists, such as the selective modulator CORT113176. Application of CORT113176 in a rat model of Alzheimer's disease showed improvement of some cognitive deficits, which was not observed with MIF[21]. Partial agonism or selective antagonism arguably have advantages in clinical settings, as they potentially allow for disease-tailored intervention in the function of the ubiquitously expressed and vital GR. Yet, they can—at the same time—also complicate the interpretation of obtained results. An alternative yet unexplored solution to overcome above mentioned obstacles would be sustained depletion of GR at the protein level. Such depletion could circumvent partial agonism and inverse agonism of some selective inhibitors, avoid crosstalk with other types of receptors and prevent adaptations often encountered in genetic deletion approaches.

In the current study, we made use of proteolysis targeting chimeras (PROTAC) technology to deplete GR protein via the cell's internal proteasome machinery. PROTACs, a special class of small molecules with promising translational potential for drug development, enable direct manipulation at the protein level by selectively inducing protein degradation[22,23]. They can be designed for any target protein to effectively induce degradation at nanomolar dosage by hijacking the ubiquitin-proteasome system (UPS). PROTACs perpetuate the degradation signal from one target protein to the next without being consumed in the process themselves. Hence, they provide an attractive approach to modulating the levels of target proteins via an event-driven mode of action rather than inhibition. Due to the low dose requirements, these drugs have a high safety profile, making them interesting candidates for translational approaches[24].

Here, we describe the successful development of a cereblon (CRBN)-recruiting PROTAC directed against the GR, and we compare its performance to existing pharmacological inhibitors in various model systems. Our results provide an in-depth functional characterization of a potent synthetic small-molecule for GR-degradation and demonstrate excellent performance in passively preventing ligand-induced gene expression activation in the absence of partial agonistic transcriptional triggering and crosstalk inhibition, allowing a straightforward interpretation of treatment outcomes. Lastly, we showcase a proof-of-concept application for targeted in vivo GR depletion without the need for laborious and time-consuming inter-crosses of genetically modified mouse lines and, hence, with direct relevance to disease models and translation potential for clinical use.

## Results

### GR-PROTAC design and linker optimization

To identify a CRBN-recruiting GR-PROTAC, we employed dexamethasone (DEX) as a small-molecule ligand for GR and lenalidomide for CRBN recruitment. By analyzing the crystal structure of DEX complexed with GR (protein data bank (PDB): 4UDC), we assessed the exit vector on the deeply embedded DEX and recognized a bile acid co-crystallized on the surface of GR at the entrance of the substrate-binding pocket (Fig. 1a)[25]. We hypothesized the bile acid could serve as a surrogate for a CRBN-recruiting ligand.

Thus, the minimal linker length required for a functional GR-PROTAC could be anticipated by measuring the distance between the DEX exit vector and the bile acid on the surface. According to the measured distances, we set out and synthesized DEX-conjugated analogs with lenalidomide and varying linker lengths and composition (Fig. 1b). Four candidates were synthesized, two of which consisted of poly(ethylene glycol) (PEG) linkers (KH-95 and KH-99), while the other two were composed of alkyl linkers (KH-103 and KH-102) (Fig. 1c and Supplementary Fig. 1, 2).

Initial treatments of the human embryonic kidney 293 (HEK293) cells with 1 µM of the candidates revealed that KH-102 and KH-103 led to efficient and rapid degradation of GR cells within less than an hour, while KH-95 and KH-99 did not induce any degradation as determined by western blot (Fig. 1d, e). Roughly half of the GR proteins were depleted within the first hour of treatment, reaching an almost complete GR depletion from 16 hours (h) onwards. Since both GR-PROTACs connected by PEG linkers (KH-95 and KH-99) were inactive, linker composition showed to be crucial for PROTAC-mediated GR degradation. Interestingly, both PROTACs based on alkyl linkers designed by our surrogate approach resulted in efficient GR-PROTACs (KH-103 and KH-102).

For a further investigation on the structure of the four PROTAC molecules in solution and their ability to form a ternary complex with GR and CRBN we performed molecular dynamics (MD) and docking simulations. Based on the gyration radii (Supplementary Fig. 3a) and distances between exit atoms (Supplementary Fig. 3b) the MD simulations revealed that KH-95 and KH-99 are less favourable (yet possible) for ternary complex formation due to shorter distances occupied by the linkers, even though a plausible docking of KH-99 was determinable using the PRosettaC web server (Supplementary Fig. 3c). KH-102 and KH-103 both in principle should allow complex formation based on the longer distance occupied by their linkers (Supplementary Fig. 3a, b) yet only KH-102 showed also positive docking (Supplementary Fig. 3d). This is in accordance with KH-102 also achieving efficient depletion (Fig. 1d, e), yet does not explain KH-103' high efficiency, which may be related to KH-103's ability to easily mimic the conformation of KH-99 (Supplementary Fig. 3f). Alternatively, the PROTACs containing alkyl linkers might show higher depletion, in comparison to a comparably long PEG linker, since PEG -substrate interactions are potentially mediated via van der Waals interaction and hydrogen bonds, as observed in another ternary complex by Gadd et al.[26].

We selected KH-103 as the most potent candidate for the treatment of cells with various concentrations of this compound and found that highly efficient degradation could still be achieved at a hundred times lower concentrations (10 nM), reaching almost full depletion at 100 nM (Fig. 1f, g). We additionally assessed the kinetics of GR degradation at 100 nM and 1 µM of KH-103 in real time using live cell confocal imaging in HEK cells transiently transfected with a plasmid encoding GFP tagged GR, that simultaneously expressed endogenous non-tagged GR, while applying a translational inhibitor to prevent protein synthesis.

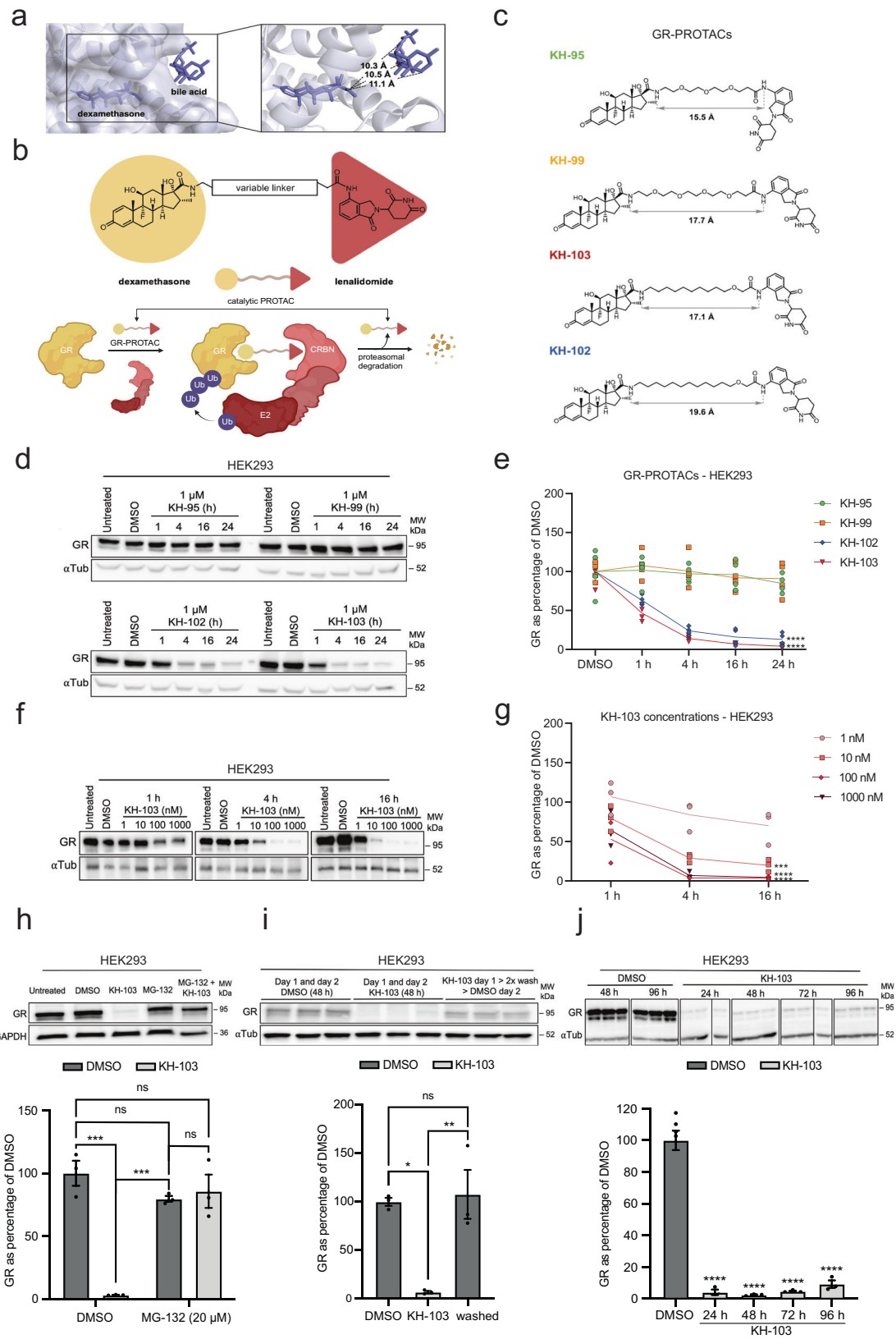

This experiment initially demonstrated nuclear translocation and subsequently achieved a complete absence of GFP signal within 4 h in the 1 µM condition and similar but slower kinetics in the 100 nM condition, while no considerable bleaching was observable in the DMSO treated control over the entire time frame (Supplementary Fig. 4).

We then tested the proteasomal dependency of the KH-103 mechanism of degradation in HEK293 cells. As expected, GR depletion by KH-103 was inhibited in the presence of 20 µM of the proteasomal inhibitor MG-132 (Fig. 1h). Additionally, the removal of KH-103 by two medium exchanges showed an almost full recovery of GR within 24 h

**Fig. 1 | PROTAC characterisation. a** The crystal structure of DEX and GR in a complex with bile acid (PDB: 4UDC). **b** Schematic representation of GR-PROTACs (upper row) and the catalytic mode of action of PROTACs (lower row). **c** GR-PROTAC candidates and their individual linker lengths. **d** Representative image and **e** quantification of immunoblots of GR in HEK293 cells following GR-PROTAC treatments ($n = 4$). (two-way ANOVA main effects for time ($F_{(4, 60)} = 34.73$, $p = <0.0001$) and for GR-PROTACs ($F_{(3, 60)} = 106.10$, $p = <0.0001$); significant interaction ($F_{(12, 60)} = 7.81$, $p = <0.0001$); follow-up Dunnett's multiple comparisons between compounds vs. DMSO at 24 h KH-102 ($p = <0.0001$) and KH-103 ($p = <0.0001$). **f** Representative immunoblot and (**g**) quantification of HEK cells treated with various concentrations of KH-103 ($n = 3$). Two-way ANOVA: significant main effects for concentration ($F_{(4, 30)} = 22.91$, $p = <0.0001$) and time ($F_{(2, 30)} = 14.43$, $p = <0.0001$). **h** Representative immunoblot and quantifications of GR degradation by KH-103 in presence of MG-132 ($n = 3$). Two-way ANOVA: significant main effects for MG-132 ($F_{(1, 8)} = 13.90$, $p = 0.0058$) and KH-103 ($F_{(1, 8)} = 29.70$, $p = 0.0006$); significant interaction ($F_{(1, 8)} = 38.20$, $p = 0.0003$). **i** Representative immunoblot and band quantification of GR in HEK293 cells following medium exchange ($n = 3$). Ordinary one-way ANOVA significant difference ($F_{(2, 6)} = 14.40$, $p = 0.0051$). Follow-up Tukey's multiple comparisons showed significant differences for DMSO vs. KH-103 ($p = 0.0102$) and KH-103 vs. washed ($p = 0.0070$). **j** Representative immunoblot and quantification of GR in HEK293 cells treated with KH-103 for up to four days. DMSO $n = 6$ (pooled 48 h and 96 h DMSO), all other conditions $n = 3$. Ordinary one-way ANOVA: significant difference ($F_{(4, 13)} = 100.00$, $p = <0.0001$). $P$-values *<0.05, **<0.01, ***<0.001, ****<0.0001. Data are presented as mean values ± SEM. Source data are provided as a Source Data file.

(Fig. 1i), indicating reversibility. When the medium was not exchanged, GR depletion was sustained up to 96 h (Fig. 1j).

## KH-103 shows efficient reversible degradation in various in-vitro models across tissue origins and species

We anticipated that KH-103 would degrade both human and rodent GR because the ligand binding site is conserved. Yet, it has been reported that PROTAC efficiency is cell-type and tissue-dependent, limited both by protein resynthesis rate and CRBN abundance[27]. Therefore, we assessed GR depletion in different cell types and across species. In the neuroblastoma mouse cell line, neuro 2a (N2a), similarly to HEK293, 100 nM of KH-103 was sufficient to induce almost complete degradation of GR (Fig. 2a–d). Likewise, we observed the same effect in a cell line expressing high levels of endogenous GR, the human lung carcinoma cell line A549 following 16 h of KH-103 treatment (Fig. 2e). N2a cells showed full maintenance of GR depletion by KH-103 up to 96 h (Fig. 2f). Also, upon medium exchange, N2a cells showed full recovery of GR protein levels (Fig. 2g). Efficient GR degradation was also achieved in mouse hippocampal and cortical primary cultures. This was confirmed by both immunoblotting (Fig. 2h) and immunofluorescence staining using a neuronal-specific marker, MAP2 (Fig. 2i), confirming KH-103's broad applicability.

## KH-103 induces nuclear translocation without triggering GR-mediated gene activation

DEX binding to GR is sufficient to cause conformational changes, which expose the nuclear translocation signal motif on the GR, causing GR translocation to the nucleus[28]. As KH-103 was designed based on the DEX binding site, we assessed whether binding of KH-103 to GR would also trigger GR translocation to the nucleus. Since our initial experiments (Fig. 1e, Supplementary Fig. 4) revealed that about half of the GR is already degraded within the first hour, we visualized the GR protein signal at various time points, starting already at 30 minutes (min) after treatment with KH-103 or DEX. Indeed, in line with immunoblot analysis, staining revealed a progressive loss of GR signal over time upon the addition of KH-103 both in HEK293 cells (Fig. 3a) and in A549 cells (Fig. 3b). DEX and KH-103 treated cells showed almost an exclusive nuclear signal for GR, indicating translocation of GR to the nucleus within 30 min. While the nuclear translocation was maintained during the 20 h of DEX treatment, the nuclear signal strongly faded over time after the addition of KH-103 (Fig. 3a, b). This observation indicated that either 1) the binding of KH-103 to GR has exposed the nuclear localization signal and induced its translocation from cytosol to the nucleus similarly to DEX, or 2) that the GR depletion occurred at different speeds in the cytosolic and nuclear UPS. To rule out that the full nuclear localization of GR was due to fast cytosolic GR degradation within 30 min, we repeated the staining in HEK293 cells at the 30 min time point in the presence of proteasome inhibitor MG-132. Again, DEX, KH-103, and KH-103 treated cells pre-incubated with MG-132 showed primarily nuclear signal for GR compared to primarily cytosolic GR signal in dimethyl sulfoxide (DMSO) and MG-132 control

conditions (Fig. 3c), confirming rapid GR translocation to the nucleus by KH-103. This finding corroborates that the rapid and high GR degradation efficiency is primarily exerted by the nuclear UPS. This feature seems especially relevant for the depletion of nuclear proteins and transcription factors by PROTACs in general.

The initial translocation of GR to the nucleus upon KH-103 raised the possibility that KH-103 treatment might trigger GR-dependent transcriptional activity. Therefore, we explored this possibility experimentally.

Following 2 h of treating A549 cells with KH-103 or DEX, we assessed the relative mRNA expression of two well-known GR targets, the immediate early genes, the serum/glucocorticoid regulated kinase 1 (Sgk1) and the dual-specificity phosphatase 1 (Dusp1). The reverse transcription-quantitative polymerase chain reaction (RT-qPCR) results showed that DEX treatment, but not KH-103 treatment, significantly increased expression (Fig. 3d)[29,30].

Moreover, co-treatment of KH-103 and DEX shows a dampened Sgk1 and Dusp1 increase as compared to DEX alone (Fig. 3d). These results point towards a potential competition between depletion and transcriptional regulation due to GR's shared binding site for its natural ligands and KH-103. On the one hand, this resulted in reduced GR activation, as observed above (Fig. 3d); on the other hand, it might also prevent efficient GR degradation. A similar competition is expected in vivo; therefore, it seemed particularly important to further assess KH-103 degradation characteristics in the co-presence of GR ligand.

## KH-103 degrades GR despite competition with DEX/cortisol binding

To test whether a putative competition would lead to decreased GR degradation efficiency, we co-treated HEK293 cells with KH-103 and with either 100 nM DEX or with a high-end physiological level (100 nM; 20x dissociation constant ($K_D$)) of the GR-ligand cortisol. We predicted less competition with cortisol due to the lower binding affinity and shorter dissociation time of cortisol from GR in comparison to DEX[31]. Immunofluorescent staining after 12 h showed almost complete degradation in co-presence with cortisol but some remaining GR protein in KH-103 co-treatment with DEX (Fig. 4a). This confirms a competition between the synthetic activating ligand DEX and KH-103. Such competition appeared negligible for the endogenous ligand cortisol/corticosterone at the tested molar ratios. Thus, GR depletion should be feasible in the presence of physiological concentrations of cortisol/corticosterone. We further assessed whether 1 h of pretreatment with DEX would compromise efficient depletion upon KH-103 addition. Prior activation of GR by DEX slightly reduces the efficiency, but importantly it does not prevent the following depletion of GR by KH-103 (Fig. 4b).

In the clinics, GR activation has proven beneficial in the treatment of multiple myeloma to suppress inflammation[32]. In associated treatment regimens DEX is often co-administered with Lenalidomide, a chemotherapeutic drug commonly used to treat multiple myeloma. Hence, we additionally assessed whether co-administration of DEX,

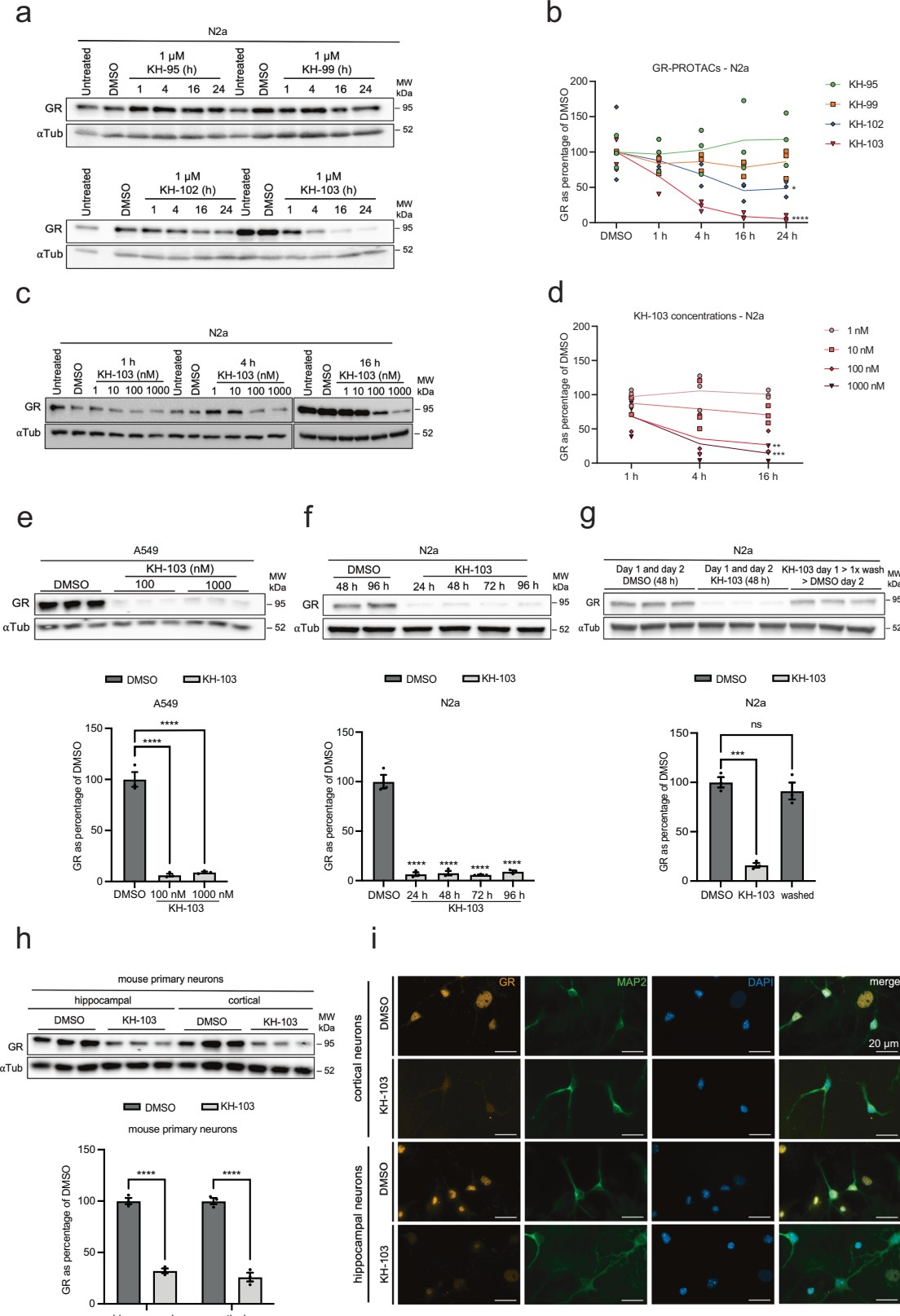

Lenalidomide and KH-103 or co-administration of Lenalidomide and KH-103 would compromise KH-103's depletion efficiency. We exposed A549 cells to either compound in isolation and or combinations for 16 h and assessed GR protein levels using immunoblotting. While Lenalidomide alone did not affect GR levels, GR depletion was as efficient in the presence of Lenalidomide as in the absence (Fig. 4c). Furthermore, DEX induces reduction of GR likely due to GR's negative transcriptional autoregulation, an effect that is not changed by the addition of Lenalidomide. We observe slightly higher remaining GR levels in the DEX + KH-103 condition as compared to the KH-103 alone pointing to the continued competition between KH-103 in the presence of DEX. This remained also unchanged by the presence of Lenalidomide.

**Fig. 2 | KH-103 induces degradation of GR in various cell types across species.**
**a** Representative immunoblot and **b** quantification of GR in N2a cells following GR-PROTAC treatments at multiple time points. Two-way ANOVA: significant main effects for time ($F_{(4,40)} = 5.21$, $p = 0.0018$) and GR-PROTACs ($F_{(3,40)} = 20.84$, $p = <0.0001$); significant interaction between them ($F_{(12,40)} = 2.51$, $p = 0.0145$). Follow-up Dunnett's multiple comparisons between compounds vs. DMSO at 24 h showed significant differences for both KH-102 ($p = 0.0382$) and KH-103 ($p = <0.0001$)). **c** Representative immunoblot and **d** quantification of comparison of treatment with KH-103 at various concentrations. Two-way ANOVA significant main effects for concentration ($F_{(4,30)} = 14.21$, $p = <0.0001$) and a trend for time ($F_{(2,30)} = 3.13$, $p = 0.0583$). Follow-up Dunnett's multiple comparison tests between various concentrations vs. 16 h time point showed significant differences for 100 nM ($p = 0.0032$) and 1000 nM ($p = 0.0006$). **e** Representative immunoblot and band quantification of KH-103 treatment in A549 cells. Ordinary one-way ANOVA: significant difference ($F_{(2,6)} = 58.8$, $p = <0.0001$). Follow-up Tukey's multiple

comparisons showed significant differences for DMSO vs. 100 nM ($p = <0.0001$) and 1000 nM ($p = <0.0001$). **f** Representative immunoblot and quantification of GR in N2a cells treated with KH-103 for up to four days (DMSO treatment for 96 h, GRs depicted as % of 48 h DMSO). Ordinary one-way ANOVA: significant difference ($F_{(4,10)} = 150.70$, $p = <0.0001$). **g** Representative immunoblot and band quantification of GR in N2a cells following medium exchange. Ordinary one-way ANOVA: significant difference ($F_{(2,6)} = 60.32$, $p = 0.0001$); Dunnett's multiple comparisons: significant difference for DMSO vs. KH-103 ($p = 0.0001$) and no significant difference between DMSO vs. washed ($p = 0.5117$). **h** Immunoblot and band quantification of GR in hippocampal and cortical primary mouse neurons. Two-way ANOVA: significant main effects for KH-103 ($F_{(1,8)} = 479.20$, $p = <0.0001$).
**i** Immunofluorescent staining of GR and MAP2, a neuronal marker, in hippocampal and cortical primary mouse neurons treated with KH-103. Scale bars are 20 μm. $N = 3$. P-values *<0.05, **<0.01, ***<0.001, ****<0.0001. Data are presented as mean values ± SEM. Source data are provided as a Source Data file.

In addition to the mere binding competition, ligand-dependent activation also induces changes in GR cellular relocalization, which could also potentially prevent GR depletion by KH-103. To test whether depletion efficiency varies according to cellular localization, we performed a fractionation experiment. Following treatment with KH-103, either in the presence or absence of DEX, fractionation yielded a cytoplasmic, nuclear, and organelle membrane-bound fraction. Qualitative comparison of compartment-specific markers, Calpain (cytosol), voltage-dependent anion channel (VDAC) (organelle-membrane), and Histone 3 (H3) (nuclear) showed enrichment of these proteins in the respective fraction, validating their identities (Fig. 4d). Immunoblotting analysis of GR levels within each fraction revealed significant interactions between DEX and KH-103 in each cellular fraction (Fig. 4e–g). Indeed, KH-103 is highly efficient at inducing a significant reduction in GR of all three cytoplasmic, organelle-bound membrane, and nuclear fractions in comparison to DMSO controls (Fig. 4e–g, conditions A vs. B). In line with immunostainings, upon co-treatment with DEX, the efficiency of KH-103 in degrading GR is reduced in all three fractions (C vs. D). Upon washing the cells after 2 h of pretreatment with DEX and incubating them with KH-103, the efficiency is recovered (Fig. 4e–g, B: vs. D: vs. F:). DEX treatment significantly reduced GR protein, both in cytosolic and membrane fractions, compared to the DMSO control, due to the expected ligand-induced translocation to the nucleus. Accordingly, GR protein significantly increased in the nuclear fraction following DEX due to the DEX-induced translocation (Fig. 4g A: vs. C:).

Interestingly, there is a portion of GR in both cytosolic and membrane fractions that, after treatment with DEX, did not translocate to the nucleus and is also not degraded by KH-103 (Fig. 4e, f C: vs. D:). The nuclear GR containing the DEX activated/translocated GR was significantly reduced when KH-103 was added (Fig. 4g C: vs. D:) implying once more that KH-103 can also degrade ligand-activated nuclear GR.

Medium wash after 2 h of DEX pretreatment decreased GR levels significantly in the nuclear fraction (Fig. 4g C: vs. E:), likely indicating relocation to the cytoplasm. Combined 16 h KH-103 and DEX treatment following 2 h DEX significantly decreased GR levels in comparison to continuous 18 h DEX treatment (Fig. 4g C: vs. D:), whereas washing out DEX after 2 h enhances depletion significantly (Fig. 4g D: vs. F:). Nevertheless, medium wash after DEX pretreatment did not fully return GR levels back to baseline (Fig. 4e A: vs. E:), indicating incomplete elimination of DEX molecules from the cells, e.g., those bound to GR. This is in agreement with the very slow DEX washout kinetics observed previously[33]. Furthermore, we observed that a portion of GR in the presence of DEX (Fig. 4e, C: vs. D:) remains in the cytoplasm and resists degradation by KH-103, raising the question of whether KH-103 might preferentially degrade some isoforms of GR while maybe having limited access to other GR isoforms. Therefore, we next investigated which GR isoforms are degradable by KH-103.

## The presence of ligand-binding domain (LBD) is required for GR isoforms to be degraded by PROTAC KH-103

The GR gene gives rise to different transcriptional isoforms with different and sometimes opposing functions (GR-α, GR-β, GR-γ, GR-A, and GR-P). They differ in their C terminal domain, which is important for ligand binding. GR-β, GR-A, and GR-P lack a full LBD. Alternative translational initiation further generates translational isoforms (GRα-A, GRα-B, GRα-C1-C3, GRα-D1-D3) with varying truncation of the N terminal domain, which is important for interaction with co-regulatory factors. Here we selected two translational isoforms, GRα-C3 and GRα-D3. Because the KH-103 design relies on binding to the LBD, we expected that only isoforms with an intact LBD would be degradable. Therefore, we tested which isoforms of GR are degradable by KH-103 by cloning the cDNA of all the splice variant isoforms as well as the two selected translational isoforms (GRα-C3 and -D3) into degradation TAG (dTAG) plasmids[34]. The experiment was performed in HEK cells, which also endogenously express GR-α, β, γ and P- isoforms (Supplementary Fig. 5) similar to A549 cells[35,36]. The expressed cDNAs harboured two HA tags and a variant of FK506-binding Protein 12 (FKBP12[F36V]) that could be targeted by the dTAG13 compound. This served as an internal control for degradability. To test whether generally additional tagging would interfere with the tertiary complex formation, we transiently expressed GR tagged with an enhanced green fluorescent protein (EGFP). Despite the large size of the tag, we still observed efficient degradation upon KH-103 treatment (Fig. 5a). Separate and transient expression of different GR isoforms in dTAG plasmids showed the correct size for all constructs (Fig. 5b, c). Moreover, dTAG13 treatment in HEK293 cells degraded all isoforms, confirming that all the tested isoforms could be targeted by a CRBN-recruiting PROTAC for ubiquitination and depletion via the proteasomal pathway. Incubation with KH-103 revealed that both C- and N-terminally tagged GR-α as well as N-terminally tagged GR-γ and C-terminally tagged GR-α-C3 and D3 showed degradation (Fig. 5d, e), consistent with the presence of a ligand binding domain in these isoforms. In contrast, KH-103 did not induce degradation of N-terminally tagged GR-β, GR-P, and GR-A isoforms, in line with the absence of an LBD in those isoforms. Interestingly, in the case of the GR-A isoform, we even observed a significant increase in GR-A levels after treatment with KH-103 in comparison to the DMSO control. Since GR-A was expressed exogenously, this could indicate that degradation of the endogenous isoforms, GR-α and GR-γ, leads to the stabilization of GR-A proteins.

## KH-103's characteristics complement current tools to study GR's genomic actions

The finding that KH-103 degrades all transcriptionally active isoforms of GR encouraged us to assess how well it can counteract DEX-induced transcriptional changes. Despite preventing agonist binding, classical small-molecule antagonists allow the binding of GR to the DNA and, to varying degrees, interactions with transcriptional co-regulators.

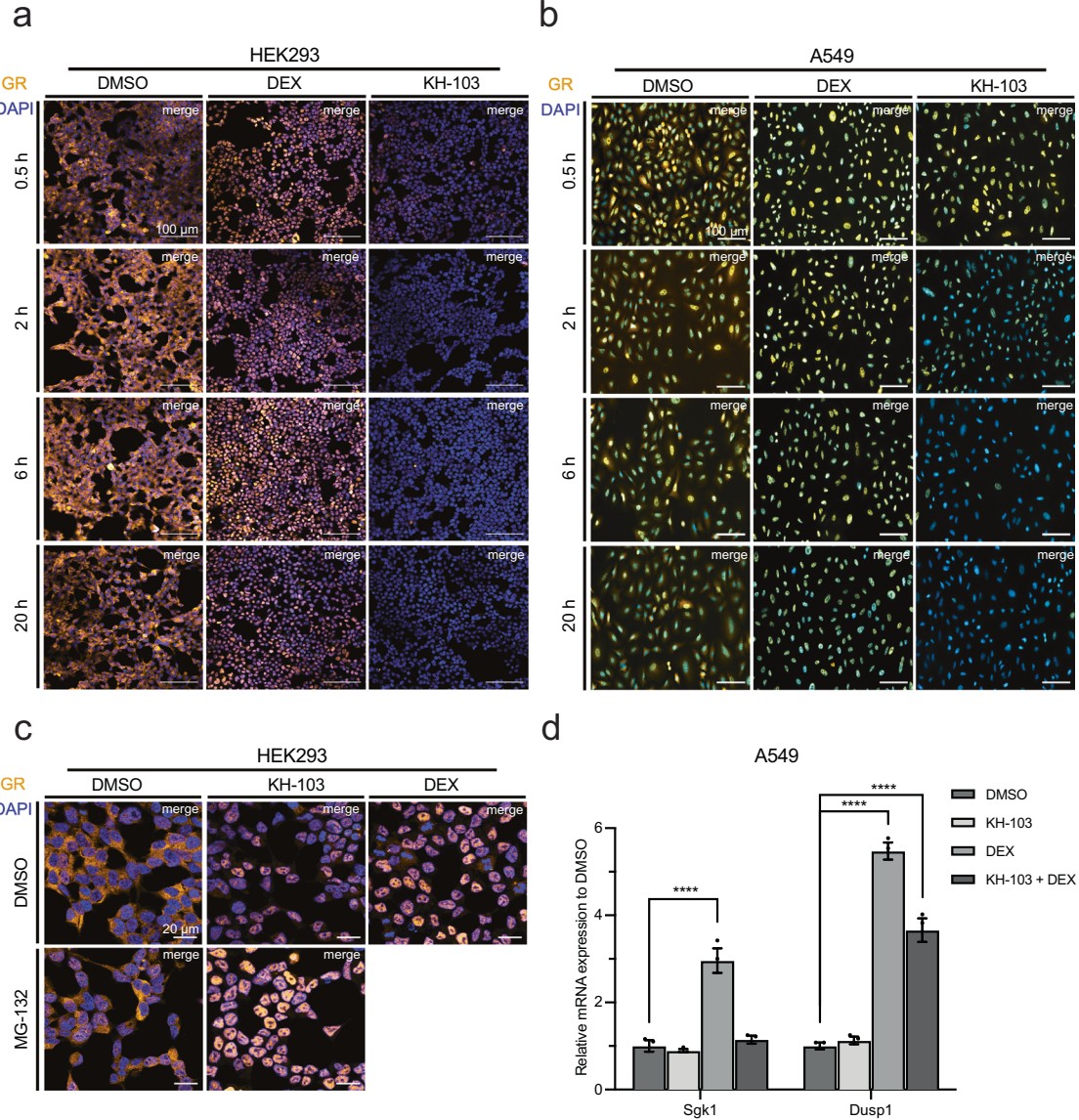

**Fig. 3 | KH-103-mediated nuclear translocation of GR.** Immunofluorescent staining of GR in HEK293 (**a**) and A549 (**b**) cells treated with DEX or KH-103 for several durations. Scale bars are 100 μm. **c** Co-treatment of KH-103 with MG-132 in HEK293 cells at 30 min. Scale bars are 20 μm. **d** Relative mRNA expression of Sgk1 and Dusp1 in A549 cells treated with DEX, or KH-103 or DEX and KH-103 for 2 h ($n = 3$). For Sgk1, two-way ANOVA showed a significant main effect for KH-103 ($F(1,8) = 34.77$, $p = 0.0004$) and DEX($F(1,8) = 45.92$, $p = 0.0001$). There was also significant interaction between KH-103 and DEX ($F(1,8) = 27.47$, $p = 0.0008$). Follow-

up Dunnett's multiple comparisons showed a significant difference between DMSO vs. DEX ($p = <0.0001$). For Dusp1, two-way ANOVA showed significant main effects for KH-103 ($F(1,8) = 22.72$, $p = 0.0014$) and DEX ($F(1,8) = 91.40$, $p = <0.0001$). There was also a significant interaction between KH-103 and DEX ($F(1,8) = 30.06$, $p = 0.0006$). Follow-up Dunnett's multiple comparisons showed significant differences between DMSO vs. DEX ($p = <0.0001$) and DMSO vs. KH-103 + DEX ($p = <0.0001$). *P*-values ****<0.0001. Data are presented as mean values ± SEM. Source data are provided as a Source Data file.

GR-PROTACs eliminate the protein and hence prevent any DNA interaction of GR and any co-regulator. To assess the functional consequences of such an alternative mode of action, we compared KH-103 to MIF, a non-selective GR antagonist with cross-reactivity with progesterone and, to a lesser extent, androgen receptors, and to CORT113176, a promising selective non-steroid GR modulator with partial agonistic actions, currently in a phase 2 trial for Amyotrophic Lateral Sclerosis (ALS)[37].

We cultured A549 cells and exposed them to different treatment regimens, including DEX (100 nM), MIF (1 μM), CORT113176 (1 μM), KH-103 (100 nM) or a combination thereof with varying exposure time and order (see treatment scheme Fig. 6a, Supplementary Fig. 6). We then assessed the consequences on gene expression using RNA sequencing. We were interested in three questions. The first question aimed at determining KH-103's potential to block DEX-induced gene expression

changes (Fig. 6a I). The second question assessed KH-103's potential to reverse DEX-induced gene expression changes (Fig. 6a II). The third question investigated the transcriptional effects of KH-103 in comparison to other inhibitors in the absence of the ligand DEX (Fig. 6a III). A scheme of the successive steps of the bioinformatic analysis is depicted in Supplementary Fig. 7)

We first confirmed that 2 h (Fig. 6a I, b, Supplementary Figs 8a–c, 9a, 10a, c) and 18 h (Fig. 6a II, c, Supplementary Fig. 8d–e, 9b, 10b, d) DEX treatment elicited significant gene expression changes. To validate our experimental setup, we next compared our DEX-induced gene expression changes (2 h and 18 h DEX treatment versus DMSO) with a publicly available dataset of 2 h and 12 h DEX treatments in the same cell line, respectively. Direction and log fold changes (logFC) showed a high correlation between both datasets (Fig. 6d, e). Furthermore, integration with

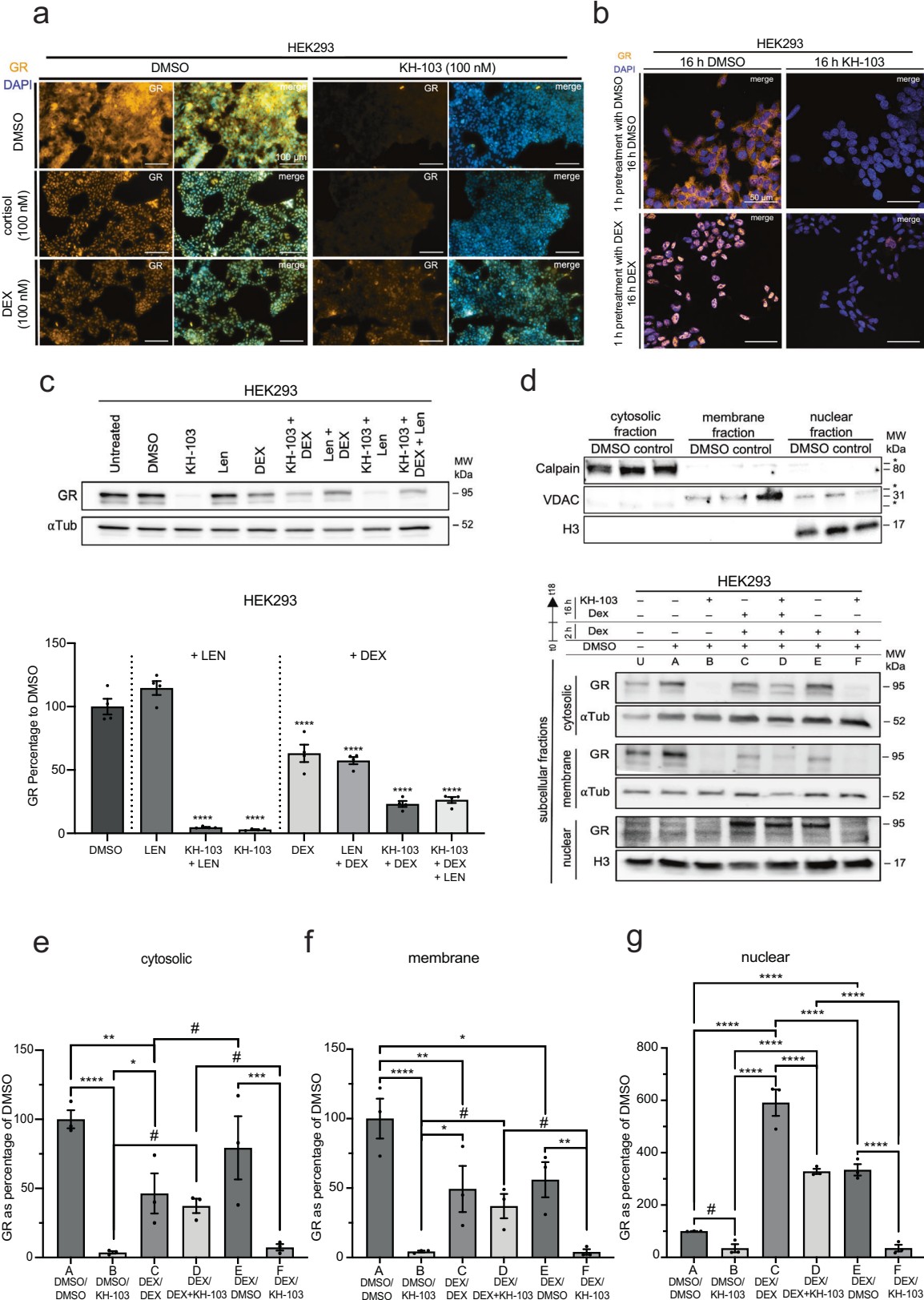

publicly available chromatin immunoprecipitation sequencing ((ChIP)-seq) data for GR showed a set of differentially expressed genes (DEGs) with clear GR peaks upon 2 h or 12 h DEX treatment, corroborating that these genes are likely directly regulated by GR (Supplementary Figs. 8, 9)[6]. Altogether, our benchmarking confirmed the reproducibility and quality of our transcriptomics data.

## KH-103 blocks DEX effects with increased efficiency compared to other inhibitors

To address question number one, the compounds' potential to block DEX-induced gene expression changes, A549 cells were first exposed to 16 h of GR modulator/degrader (MIF, CORT113176, or KH-103) or DMSO as a control, followed by 2 h of simultaneous exposure to the

**Fig. 4 | Assesment of KH-103 efficiency in co-presence with DEX.**
**a** Representative immunofluorescent staining of GR in HEK293 cells treated with KH-103, or DEX, or cortisol, or DEX + KH-103, or cortisol + KH-103. Scale bars are 100 μm. (each group $n = 3$) **b** Representative immunofluorescent staining of GR in HEK293 cells that were pretreated with DEX for 1 h followed by KH-103 treatment. Scale bars are 50 μm. **c** Representative immunoblots and quantification of GR levels upon KH-103 treatment in the absence or presence of Lenalidomide and/or DEX in HEK293 cells. All groups $n = 4$. Ordinary one-way ANOVA showed a significant difference ($F_{(7,24)} = 105.6$, $p = <0.0001$). Follow-up Dunnett's multiple comparisons showed significant differences for DMSO vs. all other groups ($p = <0.0001$) except Len which was not significant, $p = 0.0919$). Len: Lenalidomide **d** Representative immunoblot of fraction-specific markers in cytosolic, membrane, and nuclear fractions obtained from HEK293 cells. *unspecific bands. d Representative immunoblots of GR in cytosolic, membrane, and nuclear fractions obtained from HEK293 cells treated under various conditions. U: untreated, A: DMSO, B: 2 h pretreatment with DMSO followed by 16 h treatment with KH-103 (DMSO/KH-103), C: DEX/DEX,

D: DEX/DEX + KH-103, E: DEX/DMSO, F: DEX/KH-103. E and F conditions included an extra wash step after the pretreatment with DEX. **e** Band quantification of GR in the cytosolic fraction. Two-way ANOVA showed a significant main effect for KH-103 ($F_{(1, 12)} = 38.84$, $p = <0.0001$) and a significant interaction between KH-103 and DEX ($F_{(2, 12)} = 7.53$, $p = 0.0076$). There was no significant main effect for DEX ($F_{(2,12)} = 0.43$, $p = 0.6595$). **f** Band quantification of GR in the membrane fraction. Two-way ANOVA showed a significant main effect for KH-103 ($F_{(1,12)} = 35.41$, $p = <0.0001$) and a significant interaction between KH-103 and DEX ($F_{(2,12)} = 7.21$, $p = 0.0088$). There was no significant main effect for DEX ($F_{(2, 12)} = 2.06$, $p = 0.1698$). **g** Band quantification of GR in the nuclear fraction. Two-way ANOVA showed a significant main effect for KH-103 ($F_{(1,11)} = 106.30$, $p = <0.0001$) and DEX ($F_{(2, 11)} = 129.20$, $p = <0.0001$). There was also a significant interaction between KH-103 and DEX ($F_{(2,11)} = 9.88$, $p = 0.0035$). Follow-up Fisher's LSD tests for all three fractions are summarized in Supplementary Tables 1–3. $n = 3$. P-values # <0.1, *<0.05, **<0.01, ***<0.001, ****<0.0001. Data are presented as mean values ± SEM. Source data are provided as a Source Data file.

modulator/degrader and DEX (Fig. 6a I). In comparison to MIF and CORT113176, depleting GR with KH-103 prior to DEX exposure resulted in enhanced blockage of DEX-induced transcriptional changes (Fig. 7a). This result was further corroborated by hierarchical clustering, depicting that KH-103 pretreatment segregates from a cluster formed by CORT113176 and MIF, apart from the expected segregation from DEX only and the control condition (Supplementary Fig. 11a, b).

Furthermore, four k-means-based-clusters are segregated into two DEX-up-regulated and two DEX-down-regulated clusters. Additionally, this reveals slightly differential responsiveness to KH-103 across clusters, with cluster 2 being the least responsive despite more effective blockage compared to the inhibitors. Enhanced blockage by KH-103 is also evident in all other clusters but most evident in the up-regulated genes clusters: 2 and 3 (Fig. 7b). The genes up-regulated by DEX were less efficiently blocked by MIF and CORT113176 pre-incubation (clusters 2 and 3). These clusters showed more GR signal in the vicinity of their transcription start site (TSS), and about half showed GR binding at least at one putative distal enhancer[23], suggesting direct regulation via GR binding. Clusters 1 and 4 are the genes that were down-regulated upon DEX treatment, with the least responsiveness to CORT113176 in cluster 4. Despite the presence of GR signal on distal enhancers for about half of these genes when selecting the putative enhancer with the highest GR signal, we observed fewer GR signals near their TSS compared to the up-regulated genes by DEX (clusters 2 and 3). These genes are thus potentially not regulated by GR binding to their promoter (Fig. 7b). Overall, irrespective of GR signal, the clustering of up-regulated genes showed which genes are sensitive to any GR antagonists, and the clustering of down-regulated genes revealed genes sensitive to CORT113176.

Comparison of the expression levels of individual DEX-induced genes that were efficiently blocked by KH-103 with their expression upon other inhibitors pretreatment revealed 100 genes that were not blocked by CORT113176 and MIF (Supplementary Fig. 12a). In another subset of 6 DEX-induced genes, CORT113176 and MIF changed their expression significantly in the opposite direction of DEX, showing some inverse agonism, while KH-103 pretreatment efficiently prevented effects of DEX (WWTR1, MARCKS, SIX5, B4GALT5, FZD2, and IRAK2) resulting in no significant difference in expression compared to DMSO control (Supplementary Fig. 12b). This again highlights the benefit of depletion of GR at the protein level as an effective passive approach to block DEX-induced transcriptional changes.

### KH-103 shows no inverse agonism
Addressing question number two, the potential of KH-103 to reverse DEX-induced gene expression changes, cells were exposed to 2 h of DEX alone followed by 16 h of simultaneous exposure to DEX and modulator/degrader or DEX and DMSO (Fig. 6a II). Likely due to the above-described competition between DEX and KH-103 (Fig. 4,

Supplementary Fig. 6), reversal of DEX-induced changes by KH-103, after a 2 h DEX pretreatment, was overall less efficient than blocking of DEX-induced changes upon KH-103 pretreatment (Fig. 7c, d, Supplementary Fig. 13).

Nevertheless, KH-103 showed increased efficiency over the two inhibitors in the reversal of 44 genes (Supplementary Fig. 14a). Moreover, for 19 genes, KH-103 treatment reversed their expression to a level indistinguishable from DMSO, while the inhibitors led to significant overcompensation of DEX-induced changes in the opposite direction, suggesting inverse agonism (Supplementary Fig. 14b).

We further validated the efficient inhibition of some representatives of those transcripts using RT-qPCR and assessed the GR dependency of this effect with an independent approach using shRNAs directed against mRNA of GR. A549 cells were treated with DEX and either transfected with shRNA or treated with KH-103 for 16 h. Upregulation of SYBU, SEC14L2 and CITED2 mRNA was significantly prevented with both KH-103 and shRNA treatment, confirming GR dependency of this effect (Supplementary Fig. 15).

Inspection of the GR peaks at TSS revealed that DEX up-regulated genes (clusters 1 and 4) mostly showed GR ChIP-seq signal, yet not all of them showed GR at their enhancer and hence are not all conclusively direct GR targets (Fig. 7d). DEX-down-regulated genes also showed less GR signal at TSS yet some signal at a subfraction of enhancers, corroborating the findings from the above-mentioned blocking conditions that suggest indirect regulation or secondary effects of the recruitment of the transcriptional machinery to other sites. Based on the four k-means clusters, we saw both DEX-up and -down-regulated genes segregating into two clusters, one that shows more efficient reversal (cluster 3 and 4 respectively) and one reflecting less efficient reversal (cluster 1 and 2 respectively) by inhibitors, independent of GR signal at TSS or enhancers.

### KH-103 exerts passive inhibition and is specific
Our third question was whether, in the absence of ligand, GR inhibition per se is not expected to induce transcriptional changes, yet some intentional agonism of selective inhibitors respectively has been reported[17,18,38]. To address this third question, we profiled gene expression following 18 h treatment with either inhibitor or KH-103 in the absence of DEX (Fig. 8a, b). To this end, cells were exposed to 18 h of modulator/degrader (MIF or CORT113176 or KH-103) or vehicle (DMSO) (Fig. 6a III). Analyzing DEGs revealed thousands of significantly affected genes in MIF and CORT113176, consistent with partial agonistic properties. Such agonism can be explained by the unaffected DNA binding potential of inhibitor-bound GR that is absent in case of GR protein depletion. This analysis was confirmed when applying more stringent analysis criteria (|logFC| >1) (Supplementary Fig. 16). KH-103 treatment merely affected the expression of 13 genes significantly (Fig. 8a, b). This indicates a passive mode of action. Using

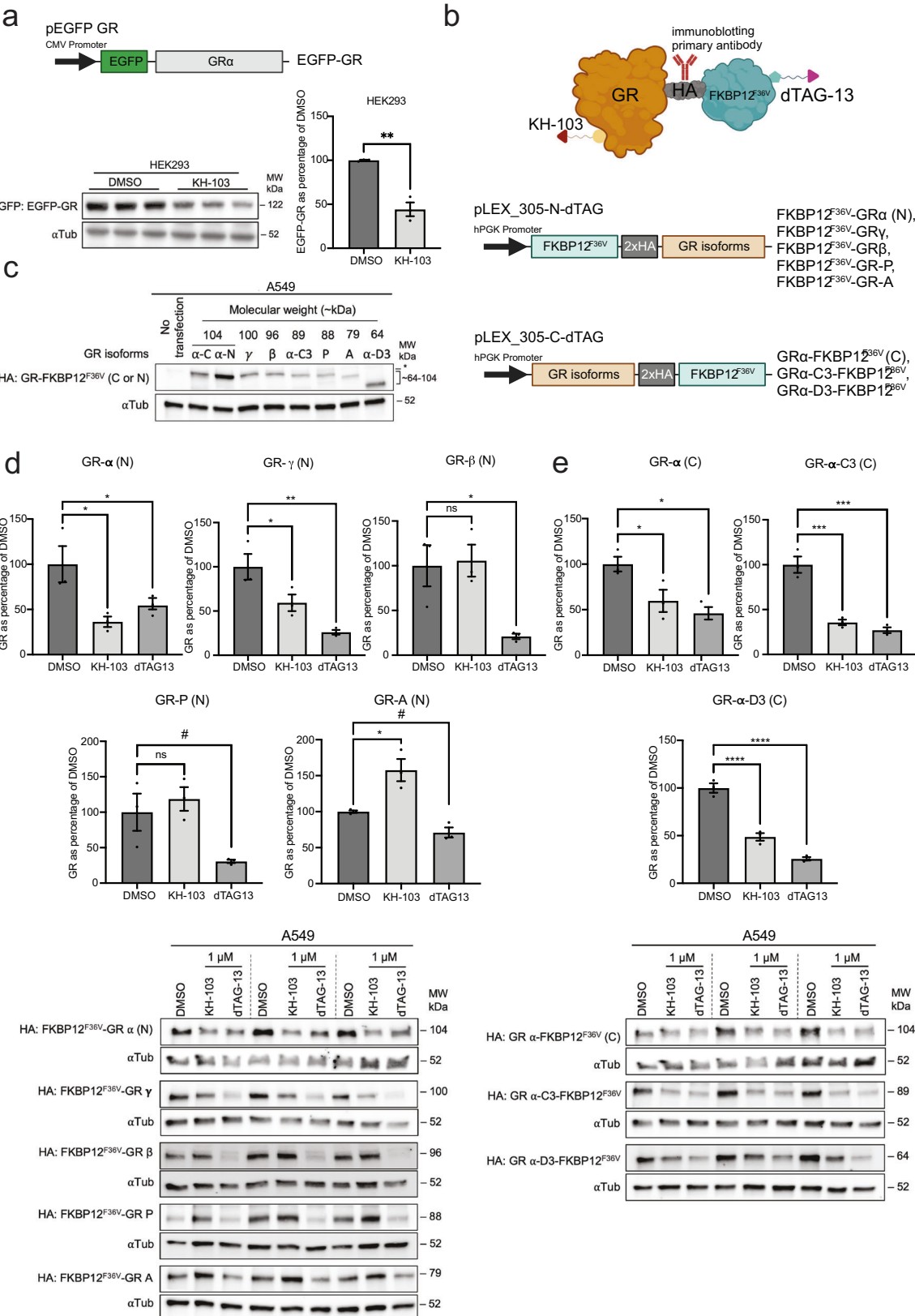

RT-qPCR, we confirmed that among those 13 DEGs in the KH-103 condition, only period circadian regulator 1 (Per1) and FK506-binding protein 51 (FKBP5) showed high fold-changes after 18 h of KH-103 treatment (Supplementary Fig. 17).

The DEX up-regulated genes in cluster 3 (Fig. 8a) were also up-regulated by MIF and by CORT113176 treatments alone. These genes

showed higher GR signals near their TSS compared to the other clusters, suggesting that paradoxically, the inhibitors partially activate GR and showed agonistic activity. The DEX down-regulated genes in cluster 1, were also down-regulated by MIF and CORT113176 but not by KH-103. These genes showed less GR signal near TSS, or enhancer compared to the up-regulated genes upon DEX (cluster 3), indicating

**Fig. 5 | GR isoforms with intact LBD are degradable by KH-103. a** Schematic depiction of plasmid used for expression of EGFP-GR fusion protein, immunoblot, and band quantification of HEK293 cells transiently expressing EGFP-GR and treated with KH-103. Unpaired two-tailed $t$ test showed a significant difference between DMSO and KH-103 ($t(4) = 7.08$, $p = 0.0021$). **b** Schematic depiction of GR isoforms in fusion with HA and FKBP12$^{F36V}$ and their expression cassettes in dTAG plasmids. These proteins were targeted for degradation upon treatment with either KH-103 or dTAG13 and were detected by HA antibody. **c** Immunoblot of transiently expressed GR isoforms in A549 cells. *: unspecific band detected by HA antibody.

**d** Band quantification and immunoblot of transcriptional GR isoforms transiently expressed in A549 cells treated with KH-103 or dTAG13. **e** Band quantification and immunoblot of translational GR isoforms transiently expressed in A549 cells treated with KH-103 or dTAG13. Results of ordinary one-way ANOVA and Holm–Sidak's multiple comparison tests for transcriptional and translational isoforms are summarized in Supplementary Table 4. $N = 3$. $P$-values $^{\#}<0.1$, $^{*}<0.05$, $^{**}<0.01$, $^{***}<0.001$, $^{****}<0.0001$. Data are presented as mean values ± SEM. Source data are provided as a Source Data file.

indirect regulation or consequence of the transcriptional machinery being recruited to other sites. The DEX-insensitive genes of cluster 2 and cluster 4 were up-regulated and down-regulated, respectively, by the inhibitors and did not show a clear TSS GR binding. This additionally points towards the indirect or off-target activity of the inhibitors (Fig. 8a, Supplementary Fig. 18a, b). Instead, most of the 13 genes triggered by KH-103 have a GR peak in their TSS or putative enhancer (Fig. 8b, Supplementary Fig. 18a, b), potentially indicating some KH-103-mediated activation, but almost no off-target effects.

Interestingly, cytochromes P450 (Cyps) genes were triggered by DEX, MIF, and CORT113176 but not by KH-103 (Supplementary Fig. 19). These genes are triggered as a response to initiate xenobiotics metabolism and are known to be regulated by the transcription factor Pregnane X receptor (PXR). This further highlights the specificity of KH-103. We verified that only 2 of the 13 KH-103 triggered genes have predicted PXR binding motifs within their TSS while also containing GR binding motifs, arguing against KH-103 triggering PXR-mediated transcriptional regulation of these genes (Fig. 8b)[39]. Overall, KH-103 shows negligible agonism or inverse agonism.

We further tested the specificity of KH-103 at the protein level in an unbiased manner using label-free proteomics in A549 cells treated with the compound for 16 h. GR was the only significantly down-regulated protein (at FDR < 0.05) among all the detected proteins. Even when applying more lenient criteria of FDR < 0.1, GR remained the sole significantly affected target that could be reliably quantified with a representation of at least 3 peptides (Fig. 8c, Supplementary Fig. 20). We then tested a putative binding partner suggested by in silico binding analysis: the secreted extracellular cytokine interleukin-6 (IL-6) and confirmed the absence of a significant change of an EGFP tagged version of IL-6 upon different dosages of KH-103 (Fig. 8d). To further explore specificity, we also assessed the structurally very similarity mineralocorticoid receptor (MR), which was not detectable in the proteomics experiment, and we also tested another putative binding partner for DEX, the SARS-Cov2 main protease (MPro))[40]. KH-103 did not induce degradation of MR at multiple concentrations tested (Fig. 8e). Since the viral enzyme, MPro, is not endogenously present in our cells, and IL-6 should be mainly extracellular, a dTAG-containing version of these proteins was transiently expressed in HEK293. Both IL-6 and MPro were depleted by the control dTAG13 PROTAC, showing that these proteins are, in principle, degradable by the PROTAC approach. Importantly, KH-103 did not induce the degradation of these proteins. Altogether these findings further support the specific targeting of GR by KH-103 (Fig. 8d–f).

## KH-103 allows interrogating the role of GR in regulating neuronal activity in primary culture

Primary cultures constitute the preferred option to study complex neuronal circuit dynamics, which are based on intercellular connectivity. Such properties cannot be interrogated in immortalized cell lines that are amenable to a genetic toolset[41]. Hence, clean pharmacological manipulations are of prime importance. To demonstrate the applicability of KH-103 in primary culture systems, we used calcium imaging in rat cortical primary neurons. It is known that GCs can shift the Ca$^{2+}$ baseline in the rat primary culture with functional

consequences on excitability[42]. The effects of prolonged GCs on spontaneous firing in primary rat cortical culture are unknown, and so is the dependency of such effect on GR.

We recorded calcium signals for 1 min at 10 Hz in four different treatment conditions. These included cultures pretreated on the first day with either KH-103 or vehicle and were subsequently supplemented with either DEX or vehicle on the second day (Fig. 9a). Measuring the spontaneous peak-to-peak intervals of the calcium signal from individual neurons, we observed that prolonged DEX treatment increased the intervals of spontaneous Ca$^{2+}$ peaks (Fig. 9b, Supplementary Fig. 21). Importantly, KH-103 alone did not influence the intervals, indicating that the mere presence of inactive GR, e.g., as a scaffold or binding partner for other proteins is irrelevant. Pretreatment with KH-103 effectively blocked the DEX-induced increase in intervals. This suggests a GR-dependent decrease in spontaneous activity in response to DEX and highlights KH-103 as an effective tool for studying the role of GR in primary culture.

## KH-103 effectively depletes GR in the pituitary in vivo and impacts corticosterone levels

Since the GR-PROTAC, KH-103 showed robust GR degradation in vitro, we aimed to assess its degradation capabilities in vivo in mice. A prime area of high GR levels and crucial for negative feedback inhibition on the HPA axis is the pituitary. We thus evaluated whether KH-103 could effectively deplete GR in the pituitary. In the first step, we intraperitoneally injected KH-103 on 2 consecutive days and assayed GR levels 4 h following the second injection using immunoblotting (Fig. 9c). We observed a significant decrease of GR (Fig. 9d), suggesting that KH-103 is also effective in depleting GR in vivo.

We then aimed at determining whether depletion of GR by KH-103 depends on the dose and administration route. To this end we subcutaneously injected KH-103 at either 0, 10, 30 or 50 mg/kg again on 2 consecutive days and harvested tissue 4 h post the second injection (Fig. 9e). We indeed found a dose-dependent effect of KH-103 on GR protein level in the pituitary as assessed via immunoblotting (Fig. 9f). Of note the depletion level of GR at the highest dose administered subcutaneously exceeded the depletion effect achieved by a similar dose via intraperitoneal injection, pinpointing the route of administration as an important determinant of efficiency.

Finally, we assessed the physiological impact of GR depletion on peripheral corticosterone levels[43]. To this end we centrally administered KH-103 twice via a stereotactically implanted bilateral cannula to the dorsal hippocampus of male mice, 12 h apart. Four hours after the second injection we subjected mice to restraint stress and assessed corticosterone levels in blood from tail pricks, at different time-points after the 30 min acute restraint stress challenge (Fig. 9g). We observed a significant interaction between time and KH-103 treatment (Fig. 9h), as well as a significant group effect of KH-103 treatment on the corticosterone dynamics following the restraint (Fig. 9i). The elevated corticosterone levels 120 min following the start of the restraint stress in the KH-103 treated group indicates a modulatory potential of hippocampal GR on the negative feedback that terminates corticosterone release after stress exposure. These results provide a proof of concept for in vivo applicability of our GR-PROTAC to assess GR-mediated physiological outcomes.

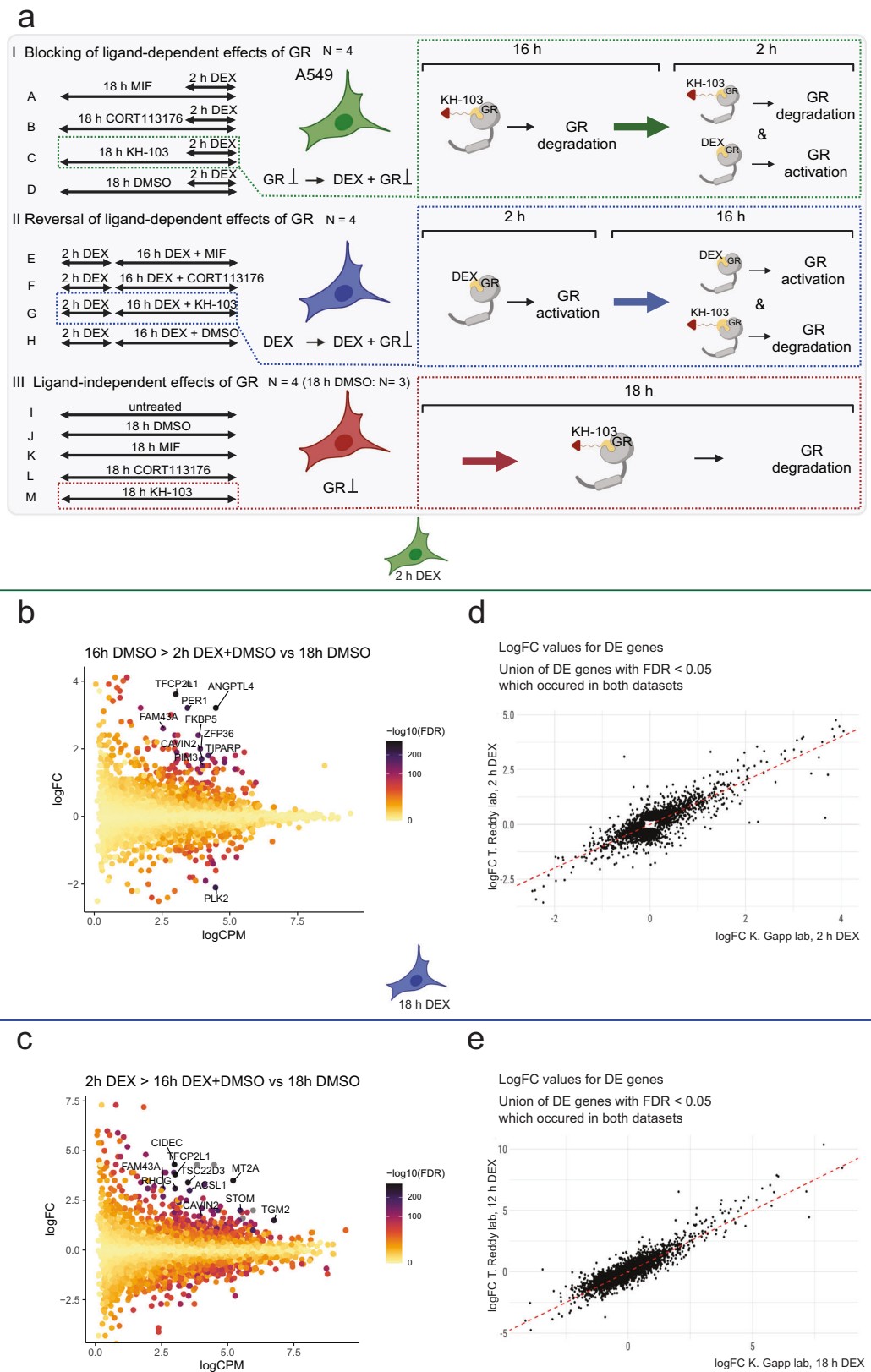

**Fig. 6 | The DEGs we identified upon 2 h, and 18 h of DEX treatments in A549 cells align with the other existing RNAseq and ChIP-seq datasets. a** Schematic depiction of the RNAseq experiment treatment groups in A549 cells. ($n \geq 3$/group). **b** MA plot of 2 h DEX DEGs (K. Gapp lab). **c** MA plot of 18 h DEX DEGs (K. Gapp lab, current study). **d** Correlation of logFC values between DEGs with the false discovery rate (FDR) < 0.05 of T. Reddy lab after 2 h DEX treatment and K. Gapp lab (current study) 2 h DEX treatment. **e** Correlation of logFC values between DEGs with FDR < 0.05 of T. Reddy lab after 12 h of DEX treatment and K. Gapp lab (current study) 18 h DEX treatment.

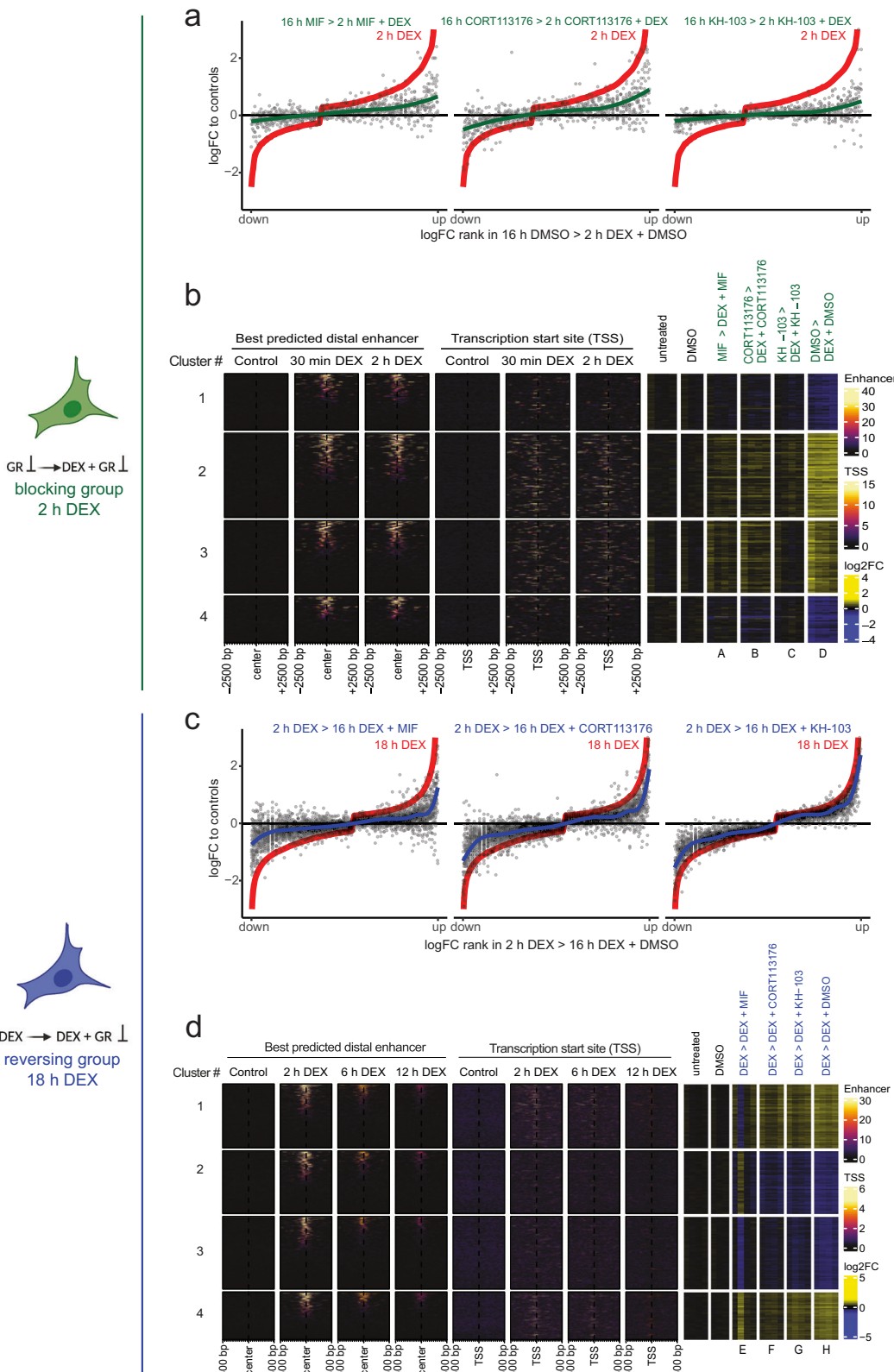

## Discussion

We developed a potentially translatable tool to abolish GR functions at the protein level. Our extensive characterization demonstrates very high efficiency of depletion that is comparable to known PROTACs against other protein targets, tested in HEK293 cells and even to PROTACs now under evaluation for clinical application directed against the androgen receptor[44,45]. Molecular dynamics and docking simulations revealed that the activity of the here described PROTACs is governed by a subtle balance between the accessible conformational space (too restricted or limited for KH-99) and favourable inter-molecular interactions (intermolecular van der Waals interactions and hydrogen bonds in case of KH-103).

KH-103 is highly specific, showing no marked cross depletion of the other tested proteins known to have the binding potential for

**Fig. 7 | KH-103 efficiently blocks DEX-induced transcriptional changes.**
**a** Comparison of logFC of DEGs upon 16 h pretreatment with CORT113176, KH-103, or MIF prior to 2 h of DEX exposure and logFC of DEGs upon no pretreatment prior to the 2 h DEX condition (red line = 2 h DEX condition). A lack of inhibition would show as the black dots following the red line, while a full inhibition would show as the black dots around the x-axis. **b** Heatmap of DEGs of the blocking treatment groups (see Fig. 6 a I) plotted aside GR binding information ± 2500 base pairs (bp) centered around their TSS, as well as ±2500 bp centered around their best predicted distal enhancer, after 30 min and 2 h DEX treatment extracted from T. Reddy ChIP-sequencing data. (n ≥ 3/group). **c** Comparison of logFC of DEGs upon 16 h treatment with inhibitors or KH-103 in co-presence of DEX after 2 h of DEX pretreatment, versus logFC of same genes upon 18 h DEX condition. **d** Heatmap of DEGs of the reversing treatment groups plotted aside GR binding information ±2500 bp centered around their TSS, as well as ±2500 bp centered around their best predicted distal enhancer, after 2 h, 6 h, and 12 h DEX treatment extracted from T. Reddy ChIP-seq data. (DEG analysis n ≥ 3/group; FDR < 0.05, |logFC| >log2(1.2)).

DEX[40]. In the absence of GR activation, KH-103 further shows neither notable agonistic nor inverse agonistic activity. In fact, activation by KH-103 seems unlikely due to the size difference between KH-103 and DEX. Should KH-103 engage in the formation of a tertiary complex, it would likely prevent the complex from binding or forming a hetero-dimer with other transcription factors. Although based on the candidate-based RT-qPCR experimental approach, we cannot exclude any fast-acting genome-wide agonistic effects, our sequencing results corroborate such interpretation. We hypothesize that this absent agonism is achieved by a complete loss of GR binding to DNA, whereas MIF still retains this ability. The same holds true for selective agonists/antagonists, which are believed to gain their selectivity by modulating specific cofactor complex formation with GR, a process that largely contributes to the context-dependent gene expression regulation via GR[46].

Jointly with the absence of agonistic activity, the high specificity contributes to the excellent passive blockage of DEX-induced gene expression changes that we observe in A495 cells (Fig. 7). In the employed tests here, KH-103 shows enhanced blockage of DEX-induced gene expression in comparison to the only clinically approved inhibitor MIF. The versatile applicability in various cellular systems and species is evidence of great potential for future use in and beyond basic research.

In a prior study, a PROTAC was developed against tubulin-associated unit (TAU) protein and showed poor blood-brain-barrier (BBB) penetration efficacy[44]. Hence, we aimed to avoid the impact of potential low brain penetration. Therefore, we first chose to assess the efficacy of depleting GR in the HPA axis outside the BBB in the pituitary, using 2 different dosing regimens, intraperitoneal and subcutaneous injections. We show that KH-103 efficiently depletes GR in either administration route. Interestingly, we observed higher depletion efficiency via the subcutaneous route, in line with known absorption differences of other drugs between different administration routes. For many drugs the maximal concentration in circulation (Cmax) reached is higher following i.p. than following s.c. injections, yet concurrently Cmax is reached later following i.p. injections[47]. Hence the optimal timepoint to observe an effect following administration likely differs between intraperitoneally and subcutaneously KH-103 injected animals. Additionally, GR has genomic and non-genomic autoregulatory action, that also modulates GR levels, potentially triggering compensatory stimulation of GR production in the pituitary[48]. Considering these complex dynamics higher depletion of GR might be expectable at a shorter time frame following i.p. administration. Future studies could investigate the detailed pharmacokinetic properties of KH-103 for in vivo application upon several administration routes. Second, we centrally administered KH-103 bilaterally into the hippocampus, a region involved in feedback on the HPA axis and the termination of the stress response[49]. This approach significantly affected GC dynamics during recovery of the restraint stress and yielded elevated GC level 90 min following the physical restraint stress, implying slight disinhibition and a role of hippocampal GR in the later phase of stress recovery. Our results are in line with the reported role of GR in mediating the genomic effects involved in negative feedback in the hippocampus[49]. This is also consistent with multiple studies showing that GR perturbation in HPA structures within the brain inhibits the negative feedback[50–54]. We speculate that at that affected timepoint the known dominance of hypothalamic and pituitary inputs on negative

feedback has ceased[43]. We cannot exclude though, that KH-103 might have diffused to regions beyond the hippocampus, including the critical PVN structures of the hypothalamus. Irrespective, the possibility of achieving depletion in the pituitary and the effective modulation of GC levels via central administration emphasize the high potency of KH-103, making us optimistic about extrapolating toward a general in vivo efficacy. Future in vivo experiments could address BBB permeability.

In terms of strategies for technology development, we demonstrate that the design of compounds can harness known ligands of proteins of interest to assemble them into PROTAC compounds, including the possibility to turn known agonists, such as DEX, into an antagonist. This strategy can be particularly helpful when the inhibition of a protein-catalytic function is not sufficient, but complete depletion is desirable. The PROTAC-mediated protein depletion approach may—in the future—also prove useful to target other steroid receptors that are involved in all sorts of pathologies.

Overall, we present our GR-PROTAC as a promising, translatable tool to achieve passive abolishment of GR's genomic function and non-genomic functions. GR-PROTAC complements the available toolbox to study GR in the context of health and disease, an endeavour that has to date, been difficult due to the cross-reactivity of full inhibitors and partial agonistic actions of many selective inhibitors. Besides benefits for basic research on steroid-mediated cellular and stress axis function, KH-103 holds great potential for clinical application in the context of stress-related neuropsychiatric disease, Cushing's disease but also for the treatment of GR-associated cancer metastasis.

## Methods

### Ethical Statement
All experiments were conducted complying with ethical regulations and local animal guidelines. The cantonal veterinary office of Zürich and the cantonal veterinary office of Basel approved our experiments: license number: ZH067/2022 and 2358.

### GR-PROTACs design and synthesis
Synthetic details on all PROTACs are provided in Supplementary Methods and Supplementary Fig. 1 and Fig. 2. NMR spectra of PROTAC KH-95, PROTAC KH-99, PROTAC KH-102, PROTAC KH-103 and their synthetic intermediates are provided in Supplementary Data 1.

### In-vitro cell culture
HEK293 and N2a cells were donated by the Schratt (ETHZ) and Mansuy (ETHZ) lab respectively and maintained in Dulbecco's Modified Eagle Medium with 10% Foetal Bovine serum (FBS) and 1% Penicillin/Streptomycin (P/S) (Gibco, Invitrogen). A549 cells (CCL-185) were purchased from American Type Culture Collection (ATCC) and were maintained in an F-12 medium with 10% FBS and 1% P/S. All cells were kept at 5% $CO_2$ at 37 °C.

### Treatment of cells with compounds
KH-95, KH-99, KH-102, KH-103, and DEX (Sigma) were used at 100 nM except indicated otherwise. MIF (Sigma-Aldrich), CORT113176 (courtesy of Corcept), and dTAG13 compounds were used at 1 µM. MG-132 (MedChemExpress) was used at 20 µM. Lenalidomide (Fluorochem) was added at a molarity of 100 nM. All mentioned compounds were dissolved in DMSO (Sigma-Aldrich, 472301).

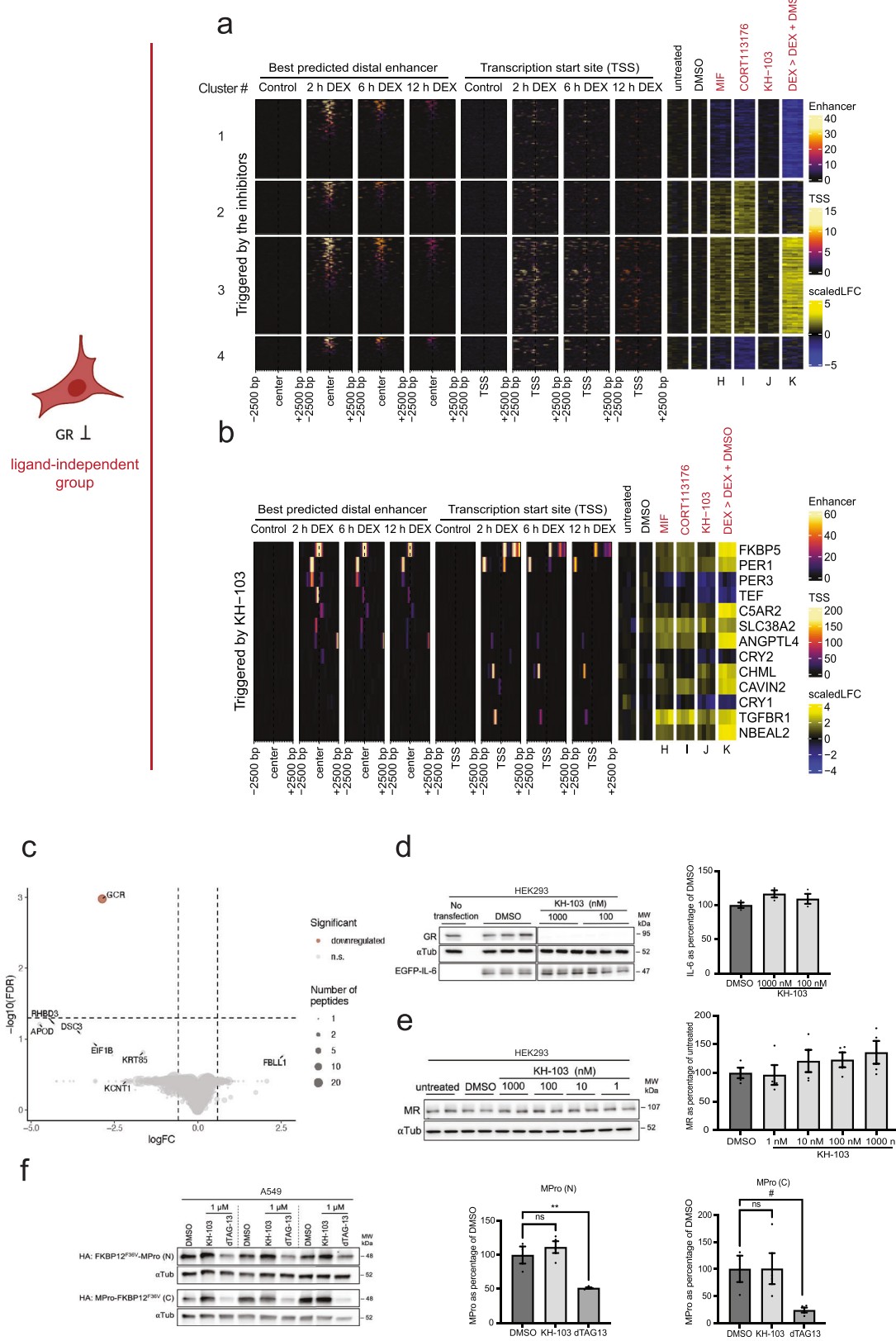

## Protein extraction and immunoblotting

Cells were lysed in RIPA buffer (Invitrogen) containing 1x Protease inhibitor (Roche) by incubating on ice for 20 min followed by 10 min centrifugation at 13 k rpm to separate protein lysate from the cell debris. Protein lysates were resolved on 10% gels (mini-PROTEAN TGX, Bio-Rad). For the fractionation experiment, the nuclear fraction proteins were resolved on gradient 8-16% gels (mini-PROTEAN TGX, Bio-Rad). Resolved proteins were transferred to nitrocellulose membranes (Trans-blot Turbo, Bio-Rad) on semi-dry Bio-Rad systems. Blots were incubated for 1 h in 5% milk-TBST and overnight at 4 °C with the primary antibody. After washes, blots were incubated with secondary antibodies for 1 h and were consequently developed using Clarity

**Fig. 8 | KH-103 treatment alone has no nonspecific transcriptional effects.**
**a** Heatmap of DEGs of the ligand-independent treatment groups plotted aside GR binding information ±2500 bp centered around their TSS, as well as ±2500 bp centered around their best predicted distal enhancer, after 2 h, 6 h, and 12 h DEX extracted from T. Reddy ChIP-seq data. (DEG analysis n ≥ 3/group; FDR < 0.05, |logFC| >log2(1.2)). **b** Heatmap of 13 DEGs triggered by KH-103 plotted aside GR binding information ±2500 bp centered around their TSS, as well as ±2500 bp centered around their best predicted distal enhancer, after 2 h, 6 h, and 12 h DEX treatment extracted from T. Reddy ChIP-seq datasets. (DEG analysis n ≥ 3/group; FDR < 0.05; |logFC| >log2(1.2)) **c** Volcano plot depicting GR as the only significantly differentially down-regulated proteins in A549 cells (n = 4/group) following 16 h 100 nM KH-103 treatment as assessed by label free proteomics. Dot size reflects number of peptides detected per protein. **d** Immunoblot and quantification of EGFP-IL6 transiently expressed in HEK293

cells treated with KH-103. N = 3. Ordinary one-way ANOVA showed no significant difference between DMSO and different KH-103 concentrations ($F_{(3, 8)} = 1.82$, $p = 0.2215$). Follow-up Dunnett's multiple comparisons tests also showed no significant differences. **e** Representative immunoblot and band quantification of MR in HEK293 cells treated with KH-103 at multiple concentrations. N = 4 independent experiments. Ordinary one-way ANOVA showed no significant difference between DMSO and different KH-103 concentrations ($F_{(4,15)} = 1.03$, $p = 0.4228$). Follow-up Dunnett's multiple comparisons tests also showed no significant differences. **f** Immunoblot and quantification of MPro fused with FKBP12$^{F36V}$ transiently expressed in A549 cells treated with KH-103 and dTAG13. For both MPro-N and -C, n = 3 for DMSO, and n = 4 for KH-103 and dTAG13 treatments. Statistical details for the MPro (N) and (C) are summarized in Supplementary Table 4. P-values $^{\#}$<0.1, **<0.01. Data are presented as mean values ±SEM. Source data are provided as a Source Data file.

Western ECL and Clarity Max Western ECL substrates (Bio-Rad). Protein bands were visualized on the ChemidocTM MP imaging system (Bio-Rad). Precision Plus Protein Dual Color Standards (catalog number 1610374) was used as the protein ladder.

## Antibodies
The following antibodies were used: GR (G-5, Santa Cruz sc-393232, 1:100), MR (clone 6G1, Merck MABS496, 1:1000), alpha-tubulin (11H10, cell signalling 2125S, dilution 1:1000), GAPDH (ABS16, Merck Millipore, 1:1000), HA (C29F4, cell signalling #3724, 1:1000), Calpain (Abcam ab28258, 1:1000), VDAC1 (Abcam ab15895, 1:1000), IL-6 (Abcam ab259341, 1:1000), H3 (Abcam ab1791, 1:1000), and GFP (Abcam, ab290, 1:1000). Secondary antibodies included: goat anti-mouse IgG antibody (Merck Millipore AP308P, (H + L) HRP conjugate, 1:20'000) and goat anti-rabbit IgG (Merck Millipore AP307P, (H + L) HRP conjugate, 1:20'000).

## Immunofluorescence staining
Cells were seeded and treated on 0.1% gelatin-coated (Pan-Biotech, P06-20410) coverslips. After a wash with 1x PBS, they were fixed in 4% PFA for 20 min at RT, followed by 3x washes in PBS before permeabilization in 0.3% Triton in PBS for 30 min at RT. After 3x further wash steps, cells were blocked overnight at 4 °C in 1% bovine serum albumin (BSA). Primary GR antibody (G-5, Santa Cruz sc-393232, 1:100) or MAP2 antibody (Chicken monoclonal anti-Map2 PA1-16751; Thermo Fisher Scientific, 1:2000) incubations lasted for 1 h at RT followed by 3x washes and 1 h incubation in secondary antibody at RT (Cy3 Goat anti-mouse against GR, Jackson ImmunoResearch 115-165-003, 1:300 and goat anti-chicken Alexa488 against MAP2, Thermo Fisher Scientific A-11039, 1:1000). After 3x washes, coverslips were mounted on glass slides using fluoroshield mounting medium plus DAPI fluorescent nuclear stain (Abcam). Slides were dried, sealed, and stored at 4 °C until imaging.

## Image acquisition−confocal/fluorescence microscopy
All immunofluorescence images were acquired using a Zeiss LSM 880 confocal microscope or Zeiss Axio observer 7 wide field inverted microscope at a specified magnification. Images were taken with ZEN 2.3 Pro software. All laser intensities and gamma values always remained the same across all conditions. Scale bars were added, and adjustments were made in ZEN 2.6 (Blue) software.

## Life confocal imaging
Glass bottom dishes (Cellvis, P12-1.5H-N) were coated with 500 µl 0,1% gelatin overnight, then HEK293 cells were seeded and grown overnight. The following day, cells were transfected with an eGFP-GR plasmid for 24 h. Prior to live-imaging, medium was removed from cells and 1 mL medium with 1uM Hoechst 33342 dye (ThermoFisher, H3570) added and incubated for 15 min. Drug solutions were prepared to be added directly prior to the start of the live-imaging after removal of the Hoecht mixture. For investigation of the degradation dynamics,

100 nM KH-103 or 1 µM KH-103 and 100 µg/mL Cycloheximide (Sigma, C4859) were added to 1 mL medium per well. As a control, 100 µg/mL Cycloheximide and 100 nM DMSO were added to 1 mL medium per well. Immunofluorescence images were acquired using a Zeiss LSM 880 confocal microscope at a ×20 magnification. Images were taken with ZEN 2.3 Pro software in intervals of 10 min in the first hour and then 15 min for a total duration of 16 h. All laser intensities and gamma values always remained the same across all conditions.

## Fractionation
Cells were seeded in 100 mm dishes. The Fractionation protocol was followed according to Baghirova S. et al., MethodsX, 2015 to acquire organelle membrane, cytosolic and nuclear fractions[55]. In brief, cells were collected by scraping and were centrifuged at 500g at 4 °C for 10 min. Pellets were washed with 1x PBS and were centrifuged again. This time pellet was lysed in ice-cold buffer A (NaCl 150 mM, HEPES (pH 7.4) 50 mM, Digitonin (Sigma, D141) 25 ug/ml, Hexylene glycol (Sigma, 112100) 1 M) containing protease inhibitor (Roche). After 10 min incubation on an end-over-end rotator at 4 °C, the lysate was centrifuged at 2000g for 10 min at 4 °C. Supernatant was collected and stored at −80 °C until further analysis as the cytosolic fraction. The pellet was lysed in ice-cold buffer B (NaCl (150 mM), HEPES (pH 7.4) 50 mM, Igepal (Sigma, I8896) 1% v:v, Hexylene glycol (Sigma, 112100) 1 M containing protease inhibitor (Roche). Lysate was incubated on ice for 30 min and subsequently centrifuged at 7000g for 10 min at 4 °C. Supernatant was collected as the membrane-bound organelle protein fraction (except those from the nucleus) and was stored at −80 °C until further analysis. Pellet was lysed in ice-cold buffer C (NaCl 150 mM, HEPES (pH 7.4) 50 mM, Sodium dodecyl sulfate (Carl Roth, CN30.3) 0.1% w:v, Hexylene glycol (Sigma, 112100) 1 M containing Benzonase (Sigma, E1014) and protease inhibitor (Roche) and was incubated on an end-over-end rotator for 30 min at 4 °C. This lysate was further sonicated for 3 × 10 s in a cold-water bath before centrifugation at 7800g for 10 min at 4 °C. Supernatant contained the nuclear protein fraction, which was stored at −80 °C until the immunoblotting analysis.

## RNA extraction
Total RNA was extracted using Quick-RNA Microprep Kit (Zymoresearch, R1051), followed by cDNA synthesis. RNA concentration was measured by NanoDrop, and integrity was checked on a 2% agarose gel.

## C-DNA synthesis
500 ng of total RNA were converted to cDNA using M-MLV Reverse Transcriptase (Promega, M1705) and Oligo(dt) 15 primers (Promega, C1101) containing RNasin® Plus RNase inhibitor (Promega, N2615) according to the manufacturer's instructions.

## RT-qPCR
cDNAs were diluted 1:5, and 2 µl of that was assessed in RT-qPCR in triplicates. HPRT and PPIA housekeeping genes were used to

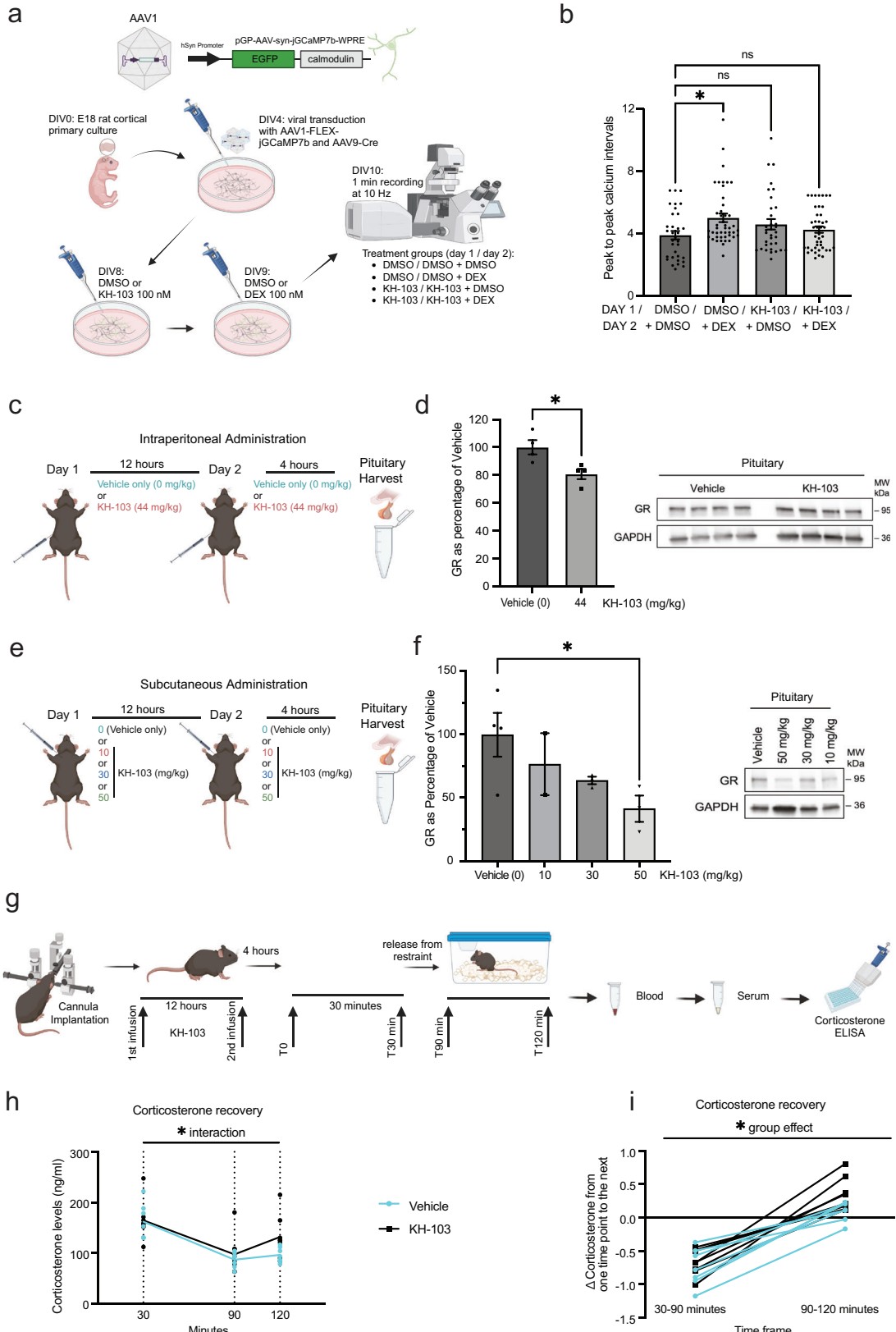

generate normalization factors using the geNorm algorithm[56]. Sybr green Mix (Roche) was used for the amplification signal, and the plates were run on a CFX384 Real-Time PCR system (Bio-Rad) device. Primers can be found in the Supplementary source data.

## RNA sequencing library preparation and sequencing

For the gene expression studies, A549 cells were treated with DMSO, KH-103, DEX, MIF, CORT113176, or combinations for varying times (please see Supplementary Fig. 3 for the immunofluorescent staining of the treatment conditions). Total RNA was isolated using the Quick-

**Fig. 9 | KH-103 application in primary neurons and in vivo. a** Schematic depiction of jGCamP7b calcium sensor cassette in AAV construct, experimental timeline and treatment groups in rat primary cortical neurons. E18: embryonic day 18, DIV: days in vitro. **b** Peak-to-peak intervals of calcium signals. Each data point (N) is the average of all intervals for each neuron during 60 s recording. n = 33 for DMSO and KH-103, n = 44 for DEX, n = 43 for DEX + KH-103. Two-way ANOVA: significant interaction between KH-103 and DEX (F(1,149) = 7.17, p = 0.0082). Follow-up Dunnett's multiple comparison tests: significant difference between DMSO and DEX (p = 0.0113). **c** Schematic depiction and timeline of KH-103 i.p. delivery and pituitary sample collection in male mice. **d** GR levels in the pituitary upon intraperitoneal injection of 44 mg/kg KH-103 (n = 4/group). Unpaired t test showed a significant effect of KH-103 t(6) = 3.016, p = 0.0235. Image of immunoblots showing GR protein in pituitary of male mice This experiment was replicated in an independent cohort of animals. **e** Schematic depiction and timeline of KH-103 subcutaneous delivery and pituitary sample collection in male mice. **f** GR levels in the pituitary upon subcutaneous injection of 0 (n = 4), 10 (n = 2), 30 (n = 3), 50 mg/kg (n = 3) KH-

103 as assessed by immunoblotting. Representative image of immunoblots showing GR protein in pituitary of male mice following subcutaneous injection of KH-103 at increasing dosages. One way ANOVA test for a linear trend: significant effect of successively increased dosage F(1,8) = 8.01, p = 0.022. Unpaired t test: significant effect of DMSO vs 50 mg/kg KH-103 t(5) = 2.623, p = 0.0469. **g** Scheme depicting timeline of stereotactic injection via implanted cannula, restraint stress paradigm and corticosterone measurements. **h** Corticosterone levels as measured in blood via ELISA 30 minutes following start of restraint, 60 min and 90 min following release. Repeated measurement ANOVA: significant interaction between time and KH-103 administration (F(2,26) = 3.57, p = 0.043, Fisher's LSD test: t(39) = 2.07, p = 0.045) (Vehicle n = 7, KH-103 n = 8). **i** Corticosterone dynamics between 30 min following start of restraint, 60 min and 90 min following release. Repeated measurement ANOVA: significant group effect of KH-103 (F(1,13) = 4.79, p = 0.048) (Vehicle n = 7, KH-103 n = 8). P-values *<0.05. **<0.01. Data are presented as mean values ± SEM. Source data are provided as a Source Data file.

---

RNA Microprep Kit (Zymoresearch, R1051). RNA samples at 25 ng/µl concentration were processed at Novogene, UK.

Non-directional Poly-A library preparations were performed according to the manufacturer's recommendations (Next® Ultra RNA Library Prep Kit for Illumina®). Briefly, mRNA was purified using poly-T oligo-attached magnetic beads, followed by fragmentation and first-strand cDNA synthesis using random hexamer primers. The second strand of cDNA was then synthesized using dTTP. This was followed by end repair, A-tailing, adapter ligation, and size selection. Next, libraries were PCR amplified and purified. For quality control and quantification, libraries were checked with Qubit and RT-qPCR. Size distribution was assessed on a bioanalyzer (Agilent 2100). Quantified libraries were then pooled. Clusters of index-coded samples were generated, and libraries were sequenced on Illumina NovaSeq 6000 S4 flowcell with PE150 to generate 6G of paired-end reads.

### Sequencing data analysis
Gene expression was quantified from RNA-seq data using Salmon 1.7.0 with the --validateMappings option on the Ensembl 105 transcriptome[57]. Counts were aggregated to gene level using the tximport package version 1.18.0[58]. For differential expression analysis, all samples were considered together, and two surrogate variables were included in the model using the sva 3.44.0 R package[59]. Differential expression analysis was performed as depicted in Supplementary Fig. 17 with edgeR 3.38.1, filtering genes using filter-ByExpr with a minimum count of 20[60]. Only differences of at least 20% (|logFC| >log2(1.2)) were considered, and the main results were replicated with a more stringent logFC threshold (Supplementary Fig. 16). For the publicly available ENCODE data, we obtained processed data from the encodeproject.org website (see GitHub repository for exact files). For RNA-seq, we used gene count tables, and for chromatin data, we used the signal p-value bigwig and the IDR-thresholded peaks across replicates. Signal heatmaps were generated with the epiwraps R package 0.99.50. To select the 'best putative distal enhancer' for given genes, we used enhancer target predictions and selected the predicted enhancer with the highest GR signal[61].

### Plasmid constructions
Codon-optimized cDNA of GR isoforms as well as MPro were synthesized by IDT and were cloned into entry plasmid Gateway™ pDONR™221 Vector (Thermo Fisher) before using LR reaction to transfer the cDNA into destination vectors pLEX_305-C-dTAG or pLEX_305-N-dTAG (Addgene #91797 & #91798, gift from James Bradner & Behnam Nabet) using gateway cloning method (Thermo Fisher)[34]. For cDNA sequences, gateway, and sequencing primers, please see Supplementary source data. Plasmid pEGFP-GR (Addgene #47504, gift from Alice Wong) was used to express GR-EGFP, and pEGFP-N1-IL6 (Addgene #111933, gift from Geert van den Bogaart)

plasmid was used for expression of IL-6-EGFP[62]. The absence of mutations in the inserted cDNAs was verified prior to use.

### Transfection
Cells were transfected at 90% confluency with the desired plasmid constructs. All transfections were done using 2 µl of P3000 and 3 µl of Lipofectamine 3000 (ThermoFisher, L3000015) per 1 µg plasmid DNA. Lipofectamine 3000 was prepared separately from the DNA and P3000 mixture in OptiMEM (ThermoFisher, 31985062). Both mixtures were combined and incubated for 15 min at RT before applying to the cells with fresh medium. For 6-well plates, a total volume of 200 µl Lipofectamine-DNA-P3000 mixture was added to each well. Mixture volumes were adjusted accordingly for smaller wells.

### shRNA treatment for GR knockdown
A549 cells were transfected for 16 h with an shRNA plasmid directed against GR (Santa Cruz sc-35505-SH) or with KH-103, DEX or DMSO co-administered with a control pmaxGFP (Lonza) to correct for potential effects resulting from the introduction of a plasmid into the cells. The medium was then changed to a treatment mix containing either DEX or DMSO for 2 h before cell harvest to mimic the treatment regime used in RNA sequencing experiments with the blocking conditions. Treatment regimens were as follow: shRNA plasmid (16 h) → DEX (2 h); pGFP + KH-103 (16 h) → DEX (2 h); pGFP + DMSO (16 h) → DMSO control (2 h); pGFP + DMSO (16 h) → DEX (2 h).

### Corticosteroid measurement
Mice were single housed and fasted with available water for 4 h. Subsequently, mice entered the acute stress response test exactly at 14:00 (t0). Mice were restrained in 50 ml Falcon tubes with 2 holes on either end for air supply and protrusion of the tail. Blood sampling was performed from tail prick at 0, 15, 30, 90 and 120 min. Blood samples were kept to naturally clot overnight at 4 °C and next day, the serum was collected by centrifugation of the samples at 2500g for 10 min at 4 °C. Serum was stored at −80 °C until analysis with ELISA. Corticosteroid levels (ng/ml) were measured in technical duplicates in 1:100 diluted samples using corticosterone Competitive ELISA Kit (Invitrogen, 15805901) and compared against the standards provided by the kit which were included on the same plate as the samples.

### Neuronal primary culture
Hippocampi or cortices of E18 pups extracted from pregnant Wistar rats (purchased from Janvier), were isolated and dissociated in 37 °C TryPLE for 7 min with inverting. TryPLE was then removed by twice washing the digested tissue in the dissection medium (Leibovitz's L-15 Medium (Gibco, 11415-04) + 7 mM HEPES (Milian, L0180-500)). Tissues were then fully dissociated by applying mechanical force using a pipette in warm Neurobasal plus (NB +) medium (Thermo Fisher,

A3582901) supplemented with 2 mM GlutaMAX, 2% B27, 1% P/S (Gibco, Invitrogen). Neurons were then counted and seeded on PLL-coated coverslips and maintained in NB+ medium in 5% $CO_2$ at 37 °C. Half of the media was refreshed every third day.

## Calcium imaging and analysis
E18 rat primary cortical neurons were cultured on polyethyleneimine (Sigma-Aldrich) and laminin (Sigma-Aldrich) coated coverslips. Cells were transduced with floxed jGCaMP7b (adeno-associated viruses (AAV) 1-syn-FLEX-jGCaMP7b-WPRE; Addgene #104493, MOI = 5 × 10⁵ vg, a gift from Douglas Kim & GENIE Project) and the same titer of Cre (AAV9-hSyn-Cre-WPRE-hGH; Addgene #105553, MOI = 5 × 10⁵ vg, a gift from James M. Wilson) AAVs at days in-vitro (DIV) 4[63]. Since this virus expresses CRE under a neuron-specific promoter, signal acquisition was restricted to neurons. The sensor used has a half decay time per 10 action potentials of ~850 ms and a half rise time per 10 action potentials of ~80 ms. Cells were treated with compounds on DIV 8 and 9. Calcium signals were recorded on DIV 10 using a Nikon NiE upright microscope equipped with Yokogawa W1 spinning disk scan head for 1 min at 10 Hz. This frame rate allows measuring at least 1 frame during the rise and a few frames during the decay of the Calcium signal with the given sensor. Calcium traces were extracted using the software Suite2p[64].

## LC-MS/MS sample preparation
To each sample (total of $n = 6$, 3/group) 100 µl of lysis buffer (4% Sodium dodecyl sulfate (SDS) in 100 mM Tris/HCl pH 8.2) were added. Protein extraction was carried out using a tissue homogenizer (TissueLyser II, QIAGEN) by applying 2 × 2 min cycles at 30 Hz. The samples were boiled at 95 °C for 10 min while shaking at 800 rpm on a Thermoshaker (Eppendorf) and treated with High Intensity Focused Ultrasound (HIFU) for 1 min at an ultrasonic amplitude of 100% before centrifugation at 20,000$g$ for 10 min. Protein concentration was determined using the Lunatic UV/Vis polychromatic spectrophotometer (Unchained Labs). The amount corresponding to 50 µg of protein was reduced with 5 mM Dithiothreitol for 30 min at room temperature followed by alkylation with 15 mM iodoacetamide at 50 °C for 30 min in the dark. Samples were processed using the single-pot solid-phase enhanced sample preparation (SP3)[65]. In short, protein purification, digest and peptide clean-up were performed using a KingFisher Flex System (Thermo Fisher Scientific) and Carboxylate-Modified Magnetic Particles (GE Life Sciences; GE65152105050250, GE45152105050250). Beads were conditioned following the manufacturer's instructions, consisting of 3 washes with water at a concentration of 1 µg/µl. Samples were diluted with 100% ethanol to a final concentration of 60% ethanol. The beads, wash solutions and samples were loaded into 96 deep well- or micro-plates and transferred to the KingFisher. Following steps were carried out on the robot: collection of beads from the last wash, protein binding to beads, washing of beads in wash solutions 1–3 (80% ethanol), protein digestion (overnight at 37 °C with a trypsin:protein ratio of 1:50 in 50 mM Triethylammonium bicarbonate) and peptide elution from the magnetic beads using MilliQ water. The digest solution and water elution were combined and dried to completeness and re-solubilized in 20 µL of MS sample buffer (3% acetonitrile, 0.1% formic acid).

## LC-MS/MS data acquisition
LC-MS/MS analysis was performed on an Orbitrap Fusion Lumos (Thermo Scientific) equipped with a Digital PicoView source (New Objective) and coupled to an M-Class UPLC (Waters). Solvent composition of the two channels was 0.1% formic acid for channel A and 99.9% acetonitrile in 0.1% formic acid for channel B. Column temperature was 50 °C. For each sample the equivalent of 300 ng of peptides were loaded on a commercial ACQUITY UPLC M-Class Symmetry C18 Trap Column (100 Å, 5 µm, 180 µm × 20 mm, Waters)

connected to a ACQUITY UPLC M-Class HSS T3 Column (100 Å, 1.8 µm, 75 µm X 250 mm, Waters). The peptides were eluted at a flow rate of 300 nL/min. After a 3 min initial hold at 5% B, a gradient from 5 to 22% B in 80 min and 22 to 32% B in additional 10 min was applied. The column was cleaned after the run by increasing to 95% B and holding 95% B for 10 min prior to re-establishing loading condition.

Samples were measured in randomized order. For the analysis of the individual samples, the mass spectrometer was operated in data-independent mode (DIA). DIA scans covered a range from 396 to 956 $m/z$ in windows of 8 $m/z$. The resolution of the DIA windows was set to 15,000, with an AGC target value of 500,000, the maximum injection time set to 22 ms and a fixed normalized collision energy (NCE) of 33%. Each instrument cycle was completed by a full MS scan monitoring 396 to 1000 $m/z$ at a resolution of 60'000.

The mass spectrometry proteomics data were handled using the local laboratory information management system (LIMS)[66] and all relevant data have been deposited to the ProteomeXchange Consortium via the PRIDE (http://www.ebi.ac.uk/pride) partner repository see data availability statement below.

## LC-MS/MS data analysis
The acquired MS raw data were converted to mzML files using msconvert (ProteoWizard release: 3.0.23052 (0c85f26), Build date: Feb 21 2023 20:02:36) with a filter for peakPicking true =1, demultiplex-optimization = overlap_only and a mass error of 10 ppm.

The mzML files were processed for identification and quantification using FragPipe (version 18.0), MSFragger (version 3.5), and Philosopher (version 4.4.0) (Yu et al., 2022). Spectra were searched against the full reviewed human proteome (20407 entries, tax 9606−Human: GCA_000001405.28 from Ensembl−https://www.uniprot.org/proteomes/UP000005640, downloaded on the 30ᵗʰ of March 2023), concatenated to its reversed decoy database, and common protein contaminants. MSFragger-DIA mode for direct identification of peptides from DIA data was used. Strict trypsin digestion with a maximum of one missed cleavage was set. The fragment ion mass tolerance was set to 20 ppm. Carbamidomethylation of cysteine was selected as a fixed modification, while methionine oxidation and N-terminal protein acetylation were set as variable modifications. EasyPQP was used to generate a DIA-NN-compatible spectral library. Subsequent quantification was performed with DIA-NN version 1.8.2.Peptide and Protein FDR were set to 0.01.

The R package prolfqua v1.1.2 was used to analyze the differential expression and to determine group differences and false discovery rates for all quantified proteins (Wolski et al., 2023). The protein lists were filtered with a threshold of 1 log2 FC and an FDR of 0.05%. The analysis was run on the local computing infrastructure (Panse et al., 2022).

## Molecular dynamics and docking simulations
MD simulations were performed employing the semiempirical quantum mechanical GFN2-xTB method implemented in 6.4.1 version of xTB software[67]. The lowest-energy conformations from a CREST search were used as initial structures with version 2.11.1 of CREST (Pracht et al., 2020). Subsequently, 10 ns MD trajectories were generated with simulation steps of 4 fs. The simulations were performed in the NVT ensemble at a temperature of 298.15 K, using the SHAKE algorithm to constrain all bonds and the GBSA implicit solvation model (H2O). Trajectories were printed out after every 100 fs. Supplementary Fig. 3g shows that the energy for each molecule fluctuates around a relatively constant value. The effect that the first frames have a lower energy was taken into account for the analysis by truncating the first 5000 frames (0.5 ns) for analysis. The choice of a semiempirical method combined with an implicit solvent model allowed us to explore conformational changes efficiently while maintaining computational feasibility. For PROTAC KH-95, PROTAC KH-99, PROTAC KH-102 and PROTAC KH-103

systems we provide the first, the 5001st and the last frame of MD simulations in Supplementary Data 2. The docking simulations were carried out using the PRosettaC web server (Zaidman et al., 2020) which samples alternatively the protein-protein interaction and the PROTAC molecule conformational space in order to predict a ternary complex. The crystal structures of the binary complexes GR + DEX (PDB: 4UDC; (Edman et al., 2015b)) and CRBN + lenalidomide (PDB: 4TZ4) were used as input.

MD simulations of the PROTAC molecules were analysed by reading out the gyration radii (Supplementary Fig. 3a) and the distances between the exit atoms (Supplementary Fig. 3b) for the whole trajectory. Gyration radii of the four molecules (Supplementary Fig. 3a) are mostly distributed between 5-7 Å. The median values of KH-99, −102 and −103 (~5.7 Å) are comparable, while the one of −95 (~5.5 Å) is smaller, as expected by the shortest linker. However, in comparison to KH-95 and −99, KH-102 and −103 populate also conformations with gyration radii >8.5 Å. Distances between the exit atoms of the four molecules (Supplementary Fig. 3b) are mostly distributed between 5-11 Å. The median values of KH-102 and −99 are comparable (~9 Å), while the ones of −103 (~8.5 Å) and −95 (~8 Å) are smaller, reflecting the order of the maximum linker length. Again, as observed for the gyration radii, KH-102 and −103 also populate more extended conformations, namely quite extended conformations with distances of >15 Å between the ligand exit atoms. Also around a distance of 12.5 Å there are significantly more conformations populated than for KH-95 and −99.

Docking simulations employing the PRosettaC web server were successful for the PROTAC molecules KH-99 and −102. For molecules KH-95 and −103 the web server '*did not find any global docking solution*'.

In the crystal structure 4UDC the distance between the exit atom of DEX and the co-crystallized bile acid on the protein surface is 8.5 Å. Taking this as a first guess of the PROTAC linker length and comparing it to the MD results, all four PROTAC molecules would be active. The results from the docking simulation are in contradiction to the experiments, where KH-102 and −103 were found to be active, whereas KH-99 is inactive. If we look at the PROTAC structures taken from the ternary complex from the docking simulations (Supplementary Fig. 3d) it is noticeable that the molecules are in a very extended conformation with distances between the exit atoms of 19.0 Å (KH-99) and 21.6 Å (KH-102). If we now take the KH-102 structure and modify it to a KH-103 (Supplementary Fig. 3f) and compare it to the two other molecules, the KH-103 is not really different to KH-99 (exit atom distance 19.2 vs 19.0 Å). However, still the docking simulation did not work successfully for KH-103. The MD potentially explains why in reality KH-103 is active while KH-99 is inactive: KH-103 more likely populates conformations with a strongly extended linker (Supplementary Fig. 3b; distance between exit atoms >12.5 Å)−and so does KH-102−while KH-99 does not populate these conformations even though the linker length potentially allows for it. Supplementary Fig. 3g shows that the energy for each molecule fluctuates around a relatively constant value. The effect that the first frames have a lower energy was taken into account for the analysis by truncating the first 5000 frames. For PROTAC KH-95, PROTAC KH-99, PROTAC KH-102 and PROTAC KH-103 systems we provide the first, the 5001st and the last frame of MD simulations in a file 'Supplementary Data 2'.

## Animals

Male mice were housed in individually ventilated cages on an inverted 12 h light-dark cycle (lights on at 08:15 am). Housing temperature was set to 21 °C and relative humidity to 55%. Food and water were provided ad libitum. They were derived from in-house breeding colonies and acclimated for at least 1 week to the respective housing room. All experiments were conducted complying with ethical regulations and local animal guidelines. The cantonal veterinary office of Zürich and the cantonal veterinary office of Basel approved our experiments: license number: ZH067/2022 and 2358.

## Injections

To prepare the stock, KH-103 powder was solved in 40% Captisol (Selleck Chemicals) in DMSO. To prepare the injection solutions, 5% of the KH-103 in Captisol/DMSO was formulated in 20% Solutol (GLPBIO) in 0.9% sterile saline (Moltox) (w:v) solution[34,68]. The final formulation contained 5% DMSO. The mixture was well vortexed and sonicated in a heated water bath until achieving a homogeneous solution ready for injection. Animals were dosed twice 12 h apart either subcutaneously (dose response curve) or intra peritoneally (single dose). Tissue was harvested 3 h following the second injection.

## Cannula implant surgery and brain drug delivery

For experiments involving hippocampal infusions, 15 male C57BL/6 mice at the age of 2–3 months were subjected to stereotactic surgery. The mice were anaesthetised with 4% isoflurane and then placed in a stereotaxic frame with continuous anaesthesia of 2% isoflurane. For analgesia, animals received a subcutaneous injection of 5 mg/kg meloxicam and buprenorphine (0.1 mg/kg), as well as application of the local analgesics lidocaine (2 mg/kg) and bupivacaine (2 mg/kg) before and after surgery. After the skull was exposed, bregma (defined as the intersection of the coronal and sagittal suture) was located and the skull placement corrected for tilt and scaling. Bilateral holes were drilled above the hippocampus at −1.8 mm AP and ±1.5 mm ML from bregma, followed by the implantation of a bilateral guide cannula (62036, RWD Life Science) into the dorsal hippocampus (coordinates from bregma: −1.8 mm AP, ±1.5 mm ML, −1.5 mm DV). The health of all animals was monitored over the course of 3 consecutive days post-surgery.

On the day of the experiment, animals were restrained, and the guide cannula was inserted with an injector needle (62236, RWD Life Science) connected to an infusion pump (R462 Syringe Pump, RWD Life Science) via plastic tubing. Afterwards, animals were allowed to freely roam their home cage for 2 min followed by bilateral intra-hippocampal infusions of 1 µl of KH-103 (~0.5 mM or 0.0044 mg/µl) at 50 µl/min. Diffusion of KH-103 was allowed for another 2 min, before the animal was detached from the infusion setup and returned to its homecage.

## In silico modelling of PROTAC binding

See Supplementary methods.

## Statistics

All statistics, except for the RNA sequencing data analysis, were performed using the built-in statistical function in GraphPad Prism. The confidence level was consistently set to 95%, and hence $p$-value below 0.05 was considered statistically significant. All error bars are represented as the standard error of the mean. Data were assumed to be normally distributed and to have equal variation across groups. The used test and statistical parameters are specified in the figure captions or the mentioned Supplementary tables.

## Reporting summary

Further information on research design is available in the Nature Portfolio Reporting Summary linked to this article.

# Data availability

The RNA sequencing data generated in this study have been deposited at GEO under accession code: GSE229084. The proteomics data generated in this study have been deposited at PRIDE[69] under the PXD046949 (Proteomic quantification following KH-103 treatment in A549 cells).

Other types of data generated in this study including imaging, Q-RT PCR, western blot and Elisa data are provided in the Supplementary Information/Source Data file. These are also deposited at the ETH research collection https://doi.org/10.3929/ethz-b-000603617. The remaining RNA sequencing data used in this study are available in the ENCODE database under accession code ENCSR897XFT (RNAseq of A549 cells treated with dexamethasone) and ENCSR210PYP (GR ChIP-seq of A549 cells treated with dexamethasone)[6]. Source data are provided with this paper.

## Code availability

The code underlying the bioinformatic analysis, including processed data objects, is available on the following GitHub repositories: https://github.com/ETHZ-INS/MG_A549 (main differential analysis) (https://doi.org/10.5281/zenodo.10062726) and https://github.com/ETHZ-INS/Glucorticoid-protacs (downstream questions and specific figures) (https://doi.org/10.5281/zenodo.10062719). GR-ChIP-Sequencing data were accessed from the ENCODE portal[6].

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

## Acknowledgements
The authors thank Hazel Hunt at Corcept for constructive comments and for providing CORT113176 and Behnam Nabet for his advice on in vivo drug delivery formulations and for providing the dTAG constructs. This work wassupported by the Swiss State Secretariat for Education, Research and Innovation (SERI) under contract number MB22.00037. Grants held by the authors: SNF PR00P3_201543, an ETH Project Grant ETH-41 20-1K.G., and the Olga Mayenfisch and Kurt und Senta Herrmann foundation K.G. Swiss Hirnliga M.G. ZNZ- Ph.D. fellowship K.G&V.F. ETH Project Grant ETH-20 19-1, Swiss National Science Foundation (grants 310030_172889/1 and 310030_204372), Swiss 3R competence center and the Botnar Research Centre for Child Health, Multi-Investigator Project J.B. ERC Advanced Grant 694829 'neuroXscales' A.H. The authors gratefully acknowledge the Functional Genomics Center Zurich (FGCZ) of the University of Zurich and ETH Zurich, and in particular Laura Kunz and Witold Wolski, for the support on Proteomics analyses. A.Ptaszek and J.Holziner were funded by the Christian Doppler Laboratory for High-Content Structural Biology and Biotechnology, Austria. The financial support by the Austrian Federal Ministry for Digital and Economic Affairs, the National Foundation for Research, Technology and Development and the Christian Doppler Research Association is gratefully acknowledged. Illustrations were made by BioRende.

## Author contributions
K.M.H., under the supervision of E.M.C., designed and synthesized GR-PROTAC compounds, obtained NMR data, provided illustrations, and helped with results interpretation. E.M.C provided resources for PROTAC synthesis. M.G., with the help of S.M.B., K.M., and R.R., performed all the characterizations in immortal cells (immunoblotting and IFS experiments, ligand competition experiments, fractionation, GR isoform experiments, and RT-qPCR experiments). M.G. performed the mouse primary neuronal cultures, the IL6 and MPro experiments, RNAseq treatments, and RNA extraction, Proteome sample treatments. P.L.G and

D.P. performed the RNA seq analyses and generated graphs. M.G. and X.X., under the supervision of A.H., performed the calcium imaging experiment. M.G., I.I., M.K., M.P., and V.F. performed mouse studies. S.F. did shRNA and Lenalidomide experiments. M.K. performed live cell imaging. J.H. carried out structural and docking simulations under guidance of R.K. M.G. prepared illustrations and graphs. M.G. and K.G. designed the experiments, interpreted the results, and wrote the manuscript with inputs from K.M.H., P.L.G., O.M., and J.B. &. K.G. provided resources.

## Competing interests

The authors declare no competing interests.
