## [Peer Review File · Nature Communications]

Reviewers' Comments:

Reviewer #1:

Remarks to the Author:

The manuscript by Gazorpak et al is focused on the generation of novel catalytically-driven GR degrader, KH-103, based on proteolysis-targeting 28 chimera technology and evaluation of the action of this degrader on cells in vitro and in vivo. The manuscript is interesting and novel.

However, I have a concern regarding effect of KH-103 on calcium signal

Critics

1. It is not clear if pure culture of neurons was used in these experiments or co-culture of neurons and glia. Please specify.
2. If it was co-culture of neurons and astrocytes - how signal from neurones and astrocytes was separated?
3. Primary neurons were used on 10 DIV. Most of the receptors expressed by 12 DIV and all experiments which related to receptor related signal should be done not earlier than 12 DIV
4. The authors discussed and represent of the histograms of the calcium peaks and intervals between peaks. However, No single trace of the calcium signal was presented. Traces of calcium signal should be presented for all measured groups.
5. If GC and KH-103 change calcium signal (intervals between peaks) - how it change amplitude of these peaks?
6. What was a frequency of the Ca²⁺ peaks? Authors used 10 Hz for measurements. Is GCAMP7B allows to measure such a changes?

Reviewer #2:

Remarks to the Author:

General comments.

In this manuscript, the authors developed KH-103, a PROTAC that induces degradation of GR, by employing DEX as a warhead. Then they examined the outcomes of KH-103 treatment in various systems, including DEX-induced translocation of GR and downstream gene expression, and found a more potent activity to suppress GR signaling than inhibitors. KH-103 may hold great potential for future clinical application.

Although they extensively analyzed the effects of KH-103 in various systems, the data presented in this manuscript are within what we can expect with this kind of degrader molecule in light of the potent activity of KH-103 to reduce the GR protein level without agonistic activity. The data are interesting in terms of pharmacology of KH-103, however, in terms of GR biology the functional data obtained with KH-103 are premature and need to be carefully confirmed to avoid misinterpretation due to an unexpected off-target effects though they show some selectivity to GR by measuring limited proteins.

Specific comments

- 1, To show the selectivity of KH-103 on the GR level convincingly, the level of various proteins upon the treatment with KH-103 should be analyzed by a comprehensive proteomics analysis.
- 2, Pharmacological data obtained with chemical compounds alone are premature to conclude the biological function of GR. Are similar results obtained by genetic perturbation, or targeted degradation of GR with other systems such as auxin-induced degradation?
- 3, In Figure 4c-g, the authors analyzed the translocation of GR by biochemical fractionation, and suggested the relocalization of GR after removal of KH-103 based on the changes in protein levels in each fraction. However, the explanation lacks the evidence that the significant amount of GR protein is degraded and de novo synthesized during the experiment.
- 4, In Figure 4c-g, was the recovery of the marker proteins equal in each fraction even after the drug treatment?

5, In Figure 5, the result that GR isoforms containing ligand binding domain are degraded but not the ones without LBD makes sense. Which isoforms of GR are expressed in the cells and in animals used in this study?

6, In figure 9d, the bar graph seems to be from the data in Supplementary Figure 12 that are very poor in the loading of equal amount of proteins and should be improved. The display below the bar graphs is confusing.

Reviewer #3:

Remarks to the Author:

This is a very nice piece of work which identifies a new strategy to prevent glucocorticoid action, using a small molecular protac approach.

The early parts of the paper are very clear, but here I would make a few suggestions...

1 crystal structures of GR bound to the protac would be very helpful. It would be useful to identify the structural basis for the failure of the PEG linked protacs.

2 the kinetics of GR degradation would massively benefit from real time analysis using fluorophore tagged GR ideally in a system where new protein synthesis is blocked.

3 the binding kinetics of Dex vs the protacs should be determined using standard tritiated Dex as ligand, with competition using unlabelled Dex vs the protac.

The second part of the paper addresses specificity of action using RNA seq. The analysis appears sound, but the data visualisation in figs 7 and 8 makes it very hard for the reader to see anything, or draw any conclusions. Ideally MA plots are better than volcano plots, and the overlaps could be visualised better using Venn diagrams. I think the logical progression here is good, but the dip into public data relating to GR ChIP-SEQ tracks does not help much. I do wonder if there should be more effort to use the RNAseq to look for off target effects...the protac used is likely to target Icaros proteins, and so can any such signature be looked for specifically?

It is surprising that the protact has no effect on MR. Dex binds the MR, which they could demonstrate in their system, using over expressed MR, in HEK cells. Is it connected to the structural features of GR binding and differences in the LBD? Could they run some modelling to explore?

The final fig is in two parts. The upper part may show a very tiny effect of Dex in terms of calcium transients. This is very unconvincing. The second part related to GR up regulation in pituitary. Again, this is an odd output. Its not clear why more typical dex dependent changes in rodent physiology are not tested, eg anti-inflammation, or effects on liver energy metabolic pathways. Its a shame to end the paper without a robust physiological end point.

Reviewer #4:

Remarks to the Author:

The work entitled " Harnessing PROTAC technology to combat stress hormone receptor activation " by Gazorpak et al. concerns the discovery of of small-molecule-based inhibitors, novel catalytically-driven glucocorticoid receptor degrader. The study based on proteolysis-targeting chimera technology (PROTAC) which enables immediate and reversible depletion of glucocorticoid receptors. The idea is based on previous work with the estrogen receptor but is interesting due to the new target which is the GR receptor, therefore the work is in the area of searching for substances that block or modify the functions of the GR receptor original. The aim of the work is well defined. The activity of best compound KH-103 was compared to two currently available inhibitors. The effects of the compounds were measured in vitro in cell cultures and in the

pituitary. KH-103 significantly inhibited, compared to existing inhibitors, gene and protein expression caused by GR receptor agonist.

The authors demonstrated the effect of KH-103 in vitro in 3 types of cell lines (HEC-293 cells, A549 and N2a mouse neuroblastoma cell line as well as in primary neuronal cells and also in the mouse pituitary. KH-103 showed robust GR degradation in vitro but a slight, although significant effect on the GR level in the pituitary after prolonged treatment. The compound produces a clear antagonistic effect on GR expression after 2 h following administration of DEX. The studies are of special interest for basic research into the functions of glucocorticoid and GR and the stress axis functions. However, the study is preliminary and more pharmacological data (eg. dose-response curve) and more replicates are needed. The effects of KH-103 on cells such as AtT20 or human corticotrophs would also be of interest.

Lenalidomide and dexamethasone is used in preclinical models and also use in clinical trials and treatment. How in this particular cell models a mixture of these substances works. It would be useful to know how a mixture of these substances performed in the experimental models studied. How does the effect of KH-103 differ from the compounds given together?

RNAseq studies were performed in A549 cell cultures upon exposure of DEX treatment. The material for RNAseq was acquired at 3 different time points (2,12 and 18 h). Positive correlations of results of Log FC obtained in two laboratories are presented. The correlations from two different labs don't seem to be needed. Perhaps a Venn diagram would be more informative than a correlation study?

Figure 6. Fig. 6 d show the logFC correlations after 2 and Fig. 6e the correlation between the results obtained after 12 and 18 h of DEX exposure. Fig. 6b is described as a correlation and shows the Volcano plot. Fig 6 f-i is missing from this figure. This figure should be corrected.

Figure 3 shows the KH-103-mediated nuclear translocation of GR. The studies were carried out on HEC293 cells and studies on the regulation of expression of two selected genes dependent on the GR receptor on a cell line are being added. Why not in Hec293?

The RNA-Seq analysis was done with the right tools (salmon + SVA + edgeR) and looks reasonable. They use the phrase FDR in the Figs, which suggests that Multiple-Testing correction has been made.

Since edgeR was used, it should be stated which type of FDR correction was applied.

It is difficult to find the size of the groups (that is, how many replicates and which ones were used).

Clear and a correct diagram explaining the "study design" for the RNA-Seq should be presented. Perhaps more information on this subject can be found at <https://github.com/ETHZ-INS/Glucocorticoid-protacs> but the page is not available.

The filtering genes in the differential analysis was done only by removing those genes with the number of reads less than 20: "filtering genes using filterByExpr with a minimum count of 20" (the question is why 20?). However, there is nothing about filtering by logFC. The pictures also show that this was not done. The recommendation for microarrays but repeated for sequencing is that a limited trust is applied to the data when $|\logFC| < 1$ [<https://doi.org/10.1038/nbt.2957>].

Furthermore, the "volcano-plot" was used, which is misleading for high-throughput technologies where all genes are studied. P-value/q-value/FDR should only be used to cut off the statistically insignificant, but further "quantification" or other assessment should be based on a comparison of logFC and average expression. Therefore, MA-plot rather than Volcano-plot is recommended.

Some semantic issue. On line 352 they write: "Analyzing differentially expressed genes (DEGs) revealed thousands of significantly affected transcripts". In principle RNA-Seq measure the expression of each of the alternative gene transcripts, but here the analysis is performed at the gene level. Although Salmon gives expression levels for alternative transcripts as a result, they themselves write that "Counts were aggregated to gene-level using the tximport package".

Therefore, the sentence on line 352 is confusing.

The broken code page and the lack of information about the group size, i.e. what exactly the RNA-Seq experiment looked like, are critical things to complete.

It would also be required to check whether if the authors apply the $|\logFC| < 1$ filter, whether their results will not change.

Reviewer #1 (Remarks to the Author):

The manuscript by Gazorpak et al is focused on the generation of novel catalytically-driven GR degrader, KH-103, based on proteolysis-targeting 28 chimera technology and evaluation of the action of this degrader on cells in vitro and in vivo. The manuscript is interesting and novel. However, I have a concern regarding effect of KH-103 on calcium signal

Critics

1. It is not clear if pure culture of neurons was used in these experiments or co-culture of neurons and glia. Please specify.
2. If it was co-culture of neurons and astrocytes - how signal from neurons and astrocytes was separated?

Answer: We appreciate the positive evaluation of our work, and we apologise that the description of the methodology caused confusion. While our culture conditions favour neurons, we expect some proportion of astrocytes in the culture. We used an AAV with a neuron-specific promoter (Synapsin) to ensure exclusive expression of the calcium indicator in neurons and can hence restrict the signal acquisition to neurons only. This is now indicated in the methods (line 847-848).

3. Primary neurons were used on 10 DIV. Most of the receptors expressed by 12 DIV and all experiments which related to receptor related signal should be done not earlier than 12 DIV

Answer: We understand that for certain experimental conditions it might be advisable to not record signal earlier than 12 DIV. However, it is quite common to perform calcium imaging for healthy rodent neuronal cultures at early stages (before DIV 12). For example, calcium imaging in rat primary cultures has been started even already at DIV 4 (Estévez-Priego et al., 2023). Many electrophysiological experiments also show that rat primary cultures exhibit spontaneous activity starting from DIV 4, which suggests that voltage-gated calcium channels are present and functional at an early stage (Yada et al., 2017). In addition, in a paper about the same culturing medium which we used, researchers also performed calcium imaging to validate the physiological function of rodent neuronal cultures at DIV 10 (Faria-Pereira et al., 2022). There are more studies in which calcium imaging was performed in rodent neuronal cultures with jRCaMP1.2 indicators from DIV 7 on (Geng et al., 2022).

4. The authors discussed and represent of the histograms of the calcium peaks and intervals between peaks. However, no single trace of the calcium signal was presented. Traces of calcium signal should be presented for all measured groups.

Answer: We thank the reviewer for this suggestion and have now added (1) representative traces of each group (2) raster plots of different conditions that depict peaks over time of each single neuron in one row. We believe this representation gives the best impression of all conditions and cells and is better than averaging traces per group, since the focus is on the dynamics represented by peak intervals rather than the evaluation of waveform features. The corresponding figures have been added to the revised version of the Supplementary Material (Supplementary Fig. 21).

5. If GC and KH-103 change calcium signal (intervals between peaks) - how it change amplitude of these peaks?

Answer: The reviewer is raising an important point. We have also initially extracted a measure corresponding to the Ca²⁺ signal amplitude and did observe a significant change upon dosage of DEX (see Figure below, peak heights in arbitrary units; two-way ANOVA revealed a significant effect of DEX, $p=0.03$). However, we believe the peak amplitude does not necessarily provide relevant information about calcium dynamics for 2 reasons:

- a. Non-linear summation: Calcium signals do not add up linearly. Instead, they interact with each other in complex, non-linear ways. For example, a second signal arriving shortly after a first one will produce a larger response (peak height) than predicted by simply adding the two signals.*
- b. Saturation effects: There are also saturation effects. For instance, if the calcium concentration in a particular area of the cell is already high, additional calcium entering the cell may not cause the expected signal increase.*

Figure: Amplitude

6. What was a frequency of the Ca²⁺ peaks? Authors used 10 Hz for measurements. Is GCAMP7B allows to measure such a changes?

Answer: The reviewer is pointing out a relevant consideration concerning signal resolution. If the frequency of the calcium peaks would refer to the number of peaks that occur per time, it could vary across different cell firing patterns. However, the intrinsic calcium dynamics are slow compared to action potentials, with decays taking up to hundreds of milliseconds. The indicator we used (jGCaMP7b) can capture such calcium events, with a half decay time - per 10 action potentials - of ~850 ms and a half rise time - per 10 action potentials - of ~80 ms. The frame rate of 10 Hz allows for measuring - at least - 1 frame during the rising and a few frames during the decay, which is a suitable sampling rate to obtain meaningful signals. In addition, previous studies even used a higher frame rate (35 Hz) with jGCaMP7b in other experiments (Dana et al., 2019). We have now added the relevant information to the revised manuscript (line 533, main manuscript).

Reviewer #2 (Remarks to the Author):

General comments.

In this manuscript, the authors developed KH-103, a PROTAC that induces degradation of GR, by employing DEX as a warhead. Then they examined the

outcomes of KH-103 treatment in various systems, including DEX-induced translocation of GR and downstream gene expression, and found a more potent activity to suppress GR signaling than inhibitors. KH-103 may hold great potential for future clinical application.

Answer: We thank the reviewer for highlighting the great potential of our KH-103 PROTAC.

Although they extensively analyzed the effects of KH-103 in various systems, the data presented in this manuscript are within what we can expect with this kind of degrader molecule in light of the potent activity of KH-103 to reduce the GR protein level without agonistic activity. The data are interesting in terms of pharmacology of KH-103, however, in terms of GR biology the functional data obtained with KH-103 are premature and need to be carefully confirmed to avoid misinterpretation due to an unexpected off-target effects though they show some selectivity to GR by measuring limited proteins.

Answer: We'd like to emphasise that obtaining this fully functional and stable degrader is an important chemical achievement that fills an unmet gap, namely non-agonistic GR targeting. We agree with the reviewer that off-target effects must be carefully ruled out. Therefore, we have now conducted a proteome wide screen and confirmed that the one and only significantly affected protein by KH-103 treatment is the Glucocorticoid receptor, as indicated below in the answer to specific comment 1. Additionally, we have further strengthened the outperformance of KH-103 over other inhibitors in blocking several Dex induced transcriptional changes by applying a variety of more stringent bioinformatic analysis parameters, further confirmed GR-dependency of certain KH-103 blocked changes with an alternate genetic knock down approach of GR and provide further in vivo data with increased biological replicates and an alternative readout, namely corticosterone levels.

Specific comments

1, To show the selectivity of KH-103 on the GR level convincingly, the level of various proteins upon the treatment with KH-103 should be analyzed by a comprehensive proteomics analysis.

Answer: We thank the reviewer for this excellent suggestion. We have now conducted a proteomic screen in A549 cells following our KH-103 incubation at the repeatedly employed molarity of 100nM for 18 hours. We could reliably detect GR protein via various peptides and can show that GR is the only significantly downregulated protein ($n=4$; FDR $p<0.05$ see Figure XA below). Even when applying more lenient statistical criteria (FDR $p<0.1$), GR remains the only reliably detectable protein (with at least 3 peptides) that seems to be downregulated (Fig. below). These data are included and discussed in the revised manuscript (Fig 8c, line 500, main manuscript, Supplementary Fig.20)

2, Pharmacological data obtained with chemical compounds alone are premature to conclude the biological function of GR. Are similar results obtained by genetic perturbation, or targeted degradation of GR with other systems such as auxin-induced degradation?

Answer: We see the reviewer's concern and appreciate the suggestion to try alternative approaches to assess the biological effect of GR depletion. We had considered degron approaches in the past, yet a colleague from the field of GR mediated gene regulation, Prof. Tim Reddy (Duke university) informed us previously that in their hands the degron approach had not proven feasible for GR. This difficulty is in line with the strong differences in the performance of different KH versions we observe and further highlights the value of our GR-PROTAC. To still address the reviewer's concern, we hence decided to instead work towards sh-RNA mediated knockdown, knowing that this would restrict the immediate application space in terms of model systems mostly to stable cell lines in the absence of stable genetic modifications, as would have been the case for the degron system.

We based our experiment on our observation in A549 cells that KH-103 was able to block certain genes' transcriptional induction by DEX while MIF and CORT113176 failed to do so. Using again A549 cells we successfully downregulated GR protein levels in A549 cells with GR-mRNA directed shRNAs, and we concomitantly also observed the blockage of DEX induced gene regulation assessed via q-PCR, for genes Cited2, Sec14L2 and SYBU (Supplementary Fig. 15, line 447 main manuscript, and Fig. below) that had not been achieved with Mifepristone or CORT113176. These results suggest that for DEX induced gene regulation of these targets the physical presence of GR protein is required. For instance, Cited2 is regulated both via direct GR dependent

transcriptional regulation (Chinenov et al., 2014) and GR-mediated mRNA decay (Zhu et al., 2021) likely explaining why the occupancy driven way of action of the other inhibitors was insufficient to block the upregulatory effect of DEX. It showcases the importance of alternative GR manipulation tools for proteins that have varied modes of actions apart from classical transcription factor binding to DNA target sequences for regulation of mRNA levels.

Altogether, it emphasises the benefit of KH-103 when interrogating questions that are otherwise inaccessible because of insufficient depletion, compensatory effects as a result of stable genetic alteration, or toxicity in difficult to transfect systems such as primary culture.

3, In Figure 4c-g, the authors analyzed the translocation of GR by biochemical fractionation, and suggested the relocalization of GR after removal of KH-103 based on the changes in protein levels in each fraction. However, the explanation lacks the evidence that the significant amount of GR protein is degraded and de novo synthesized during the experiment.

Answer: We assume the reviewer referred to the sentence: "Nevertheless, medium wash after DEX pretreatment did not fully return GR levels back to baseline (previous Fig. 4g A: vs. E:), indicating incomplete elimination of DEX molecules from the cells, e.g., those bound to GR." In fact, we do not argue for a relocalization of GR following KH-103 removal, since the experiment does not contain any condition where KH-103 was removed. We argue for an initial conformational change in GR upon KH-103 binding, which can be regarded as an "activation" that causes GR to relocate to the nucleus. Following KH-103 application, GR seems to be located almost entirely in the nucleus. This becomes apparent if we block the proteasome pharmacologically (Fig. 3c main manuscript). It is true that it cannot be ruled out that in part, the high levels of GR in the cytosol after the removal of DEX from the medium are not due to resynthesis. However, relocation of GR to the cytosol in the absence of active depletion is well in line with the literature (Galigniana et al., 1999; Madan & DeFranco, 1993)(DeFranco et al., 1991). The strong effect on GR levels in all 3 fractions argues for a potent depletion despite a potential de novo-synthesis going on in parallel. If the reviewer can confirm that we correctly identified what he/she is referring to, we can happily amend this impression in the text.

4, In Figure 4c-g, was the recovery of the marker proteins equal in each fraction even after the drug treatment?

Answer: We apologise if our experimental description was not clear enough. Within each fraction we normalised the GR levels to the amount of endogenous control. The marker proteins were not quantified per se but served as an identification and confirmation of purity. Furthermore, in the nuclear fraction the protein used as a "marker" is identical to the housekeeping gene. We did not observe a significant difference in the total protein amount within each fraction between drug-treated and vehicle-treated sample fractions, and while we did stain for the markers the bands were not ideal for quantification against a housekeeping gene. Nevertheless, we did now quantify against aTub and did not observe a significant change for neither Calpain, nor VDAC (see Figures below). For H3 we are unable to quantify since it is used as the housekeeping gene. We have clarified our description of this experiment in the revised version (line 305, main manuscript).

Representative Immunoblot image of Cytoplasmic fraction

Representative Immunoblot image of membrane fraction

Quantifications (VDAC and Calpain normalised to aTub)

5, In Figure 5, the result that GR isoforms containing ligand binding domain are degraded but not the ones without LBD makes sense. Which isoforms of GR are expressed in the cells and in animals used in this study?

Answer: According to the literature, A549 and HEK cells express the transcriptional isoforms alpha and to a lesser extent beta GR isoforms (Morgan et al., 2016) (Pujols et al., 2012). Based on our own RNAseq experiments, we can additionally conclude that A549 cells reliably express the following isoforms: GR-Alpha and to a much lesser extent GR gamma (note the scaling of the x-axis). GR beta is not reliably detectable (Figure A below). Lack of detection of course does not exclude expression altogether, since this might be due to the sensitivity of the method and the chosen sequencing depth. Further, q-RT-PCR experiments using primers specific for isoform A, P, beta and gamma or all isoforms in A549 and HEK293 cells confirm the detection of gamma and further amplified the P isoform (Figure B, C below). We were unable to amplify/detect isoform A and isoform beta (see list of primers used below) in either cell type and did not find any isoform expression information in the literature for N2a cells and mouse hippocampal primary cells, despite extensive literature study. We have no reason to believe that the mice used in this study lack the expression of any specific isoform, yet we did not directly assess this here in a systematic way. We have added the available information to the revised manuscript (line 346 main, Supplementary Fig.5).

A) RNA sequencing results of GR transcriptional isoforms in A549 cells

B) Q-RT-PCR results of GR transcriptional isoforms in HEK cells (16 hours treatment)

C) Q-RT-PCR results of GR transcriptional isoforms in A549 cells

Primers sequences:

Forward

Reverse

qRT1_h_GR_All	GCTATTCAAGCCCCAGCAT G	CAGCTTCCACAAGTTAAGAC TC
qRT5_h_GR-Gamma	GCTATTCAAGCCCCAGCAT G	GTGCTGTCTACCTTCCACTG
qRT6_h_GR-Beta	CTCTTCAGTTCCTAAGGACG G	GATTAATGTGTGAGATGTGC TTTC

qRT7_h_GR-A	GAAGGACAGCACAATTACC TATG	CCGTCCTTAGGAACAGCTTC
qRT8_h_GR-P	GAGCAGAGAATGACTCTAC CC	CCAACCTGAAGAGAGAAGCA G

6, In figure 9d, the bar graph seems to be from the data in Supplementary Figure 12 that are very poor in the loading of equal amount of proteins and should be improved. The display below the bar graphs is confusing.

Answer: We agree with the reviewer on the importance of ideally loading equal amounts of proteins. The unequal loading is the result of the minute amounts of protein that can be retrieved from the tiny pituitary that prompted us to better load all the protein we had harvested. Even though this should be accounted for by the internal normalisation against the endogenous control, we have now repeated the in vivo KH-103 administration to confirm the downregulation of GR in pituitary upon KH-103. Indeed, we were able to achieve a much more consistent loading amount. The revised blot is now available under Supplementary Fig 23 (also attached below).

Reviewer #3 (Remarks to the Author):

This is a very nice piece of work which identifies a new strategy to prevent glucocorticoid action, using a small molecular protac approach.

Answer: We are pleased that the reviewer likes our work.

The early parts of the paper are very clear, but here I would make a few suggestions...

1 crystal structures of GR bound to the protac would be very helpful. It would be useful to identify the structural basis for the failure of the PEG linked protacs.

Answer: We thank the reviewer for this interesting suggestion that we indeed had considered before. We initiated a collaboration with Dr. Rebecca Beveridge from Strathclyde University to investigate differences in ternary complex formation between the PEG- and alkyl-base GR-PROTACs with native MS. We had unfortunately failed in ionizing GR. Alternatively, we have now teamed up with the group of Prof. Konrat who is an expert in computational modelling of interactions. Their in silico molecular dynamics simulations and docking simulations based on NMR data showed that KH-95

and KH-99 are less favourable for ternary complex formation (Supplementary Fig.3a,b, and below) due to too short distances occupied by the linkers, even though docking of KH-99 was positive (Supplementary Fig.3c and below). KH-102 and KH-103 both in principle should allow complex formation based on the longer distances occupied by their linkers (Supplementary Fig.3a,b and below) yet only KH-102 showing also positive docking (Supplementary Fig.3d and below). This is in accordance with KH-102 also showing efficient depletion (Fig.1d,e), yet does not explain KH-103 efficiency. The efficiency may be related to KH-103's ability to easily mimic the conformation of KH-99 (Supplementary Fig.3f and below). Alternatively, the PROTACs containing alkyl linkers might show higher depletion, in comparison to a comparably long PEG linker since the PEG potentially interacts through van der Waals interaction and hydrogen bonds with the substrate, as observed in another ternary complex by Gadd et al (Gadd et al., 2017). These findings are now described in the revised manuscript (line 182 main text).

Histograms of the gyration radii of the four PROTACs KH-95 (top, left), -99 (top, right), -102 (bottom, left) and -103 (bottom, right) from MD simulations. The red dashed line marks the median value.

Histograms of the distances between the exit atoms of the four PROTACs KH-95 (top, left), -99 (top, right), -102 (bottom, left) and -103 (bottom, right) from MD simulations. The red dashed line marks the median value.

Structure of the ternary complex of GR – KH-99 – CRBN (light pink – orange – light green) from two different viewing angles (top) and the corresponding clip outs (bottom) as simulated by PROsettaC and reported as the highest score.

Structure of the ternary complex of GR – KH-102 – CRBN (light pink – blue – light green) from two different viewing angles (top) and the corresponding clip outs (bottom) as simulated by PROsettaC and reported as the highest score.

Structure of the PROTAC molecules KH-99 (orange) and -102 (blue), including the distance between the exit atoms, taken from the ternary structure as simulated by PRosettaC and reported as the highest score.

Structure of the PROTAC molecules KH-99 (orange) and -102 (blue) in comparison with KH-103 (green; modified from the KH-102 structure), including the distance between the exit atoms, taken from the ternary structure as simulated by PRosettaC and reported as the highest score.

2 the kinetics of GR degradation would massively benefit from real time analysis using fluorophore tagged GR ideally in a system where new protein synthesis is blocked.

Answer: We followed the reviewer's recommendation and performed a real time analysis by life imaging of HEK cells transfected with GR-GFP in the presence and absence of the protein synthesis inhibitor, Cycloheximide. Our results clearly show an efficient degradation already in the first hours of exposure to KH_103 at 1uM and a slightly delayed response at 100nM (Supplementary Fig. 4, and Fig. attached below).

3 the binding kinetics of Dex vs the protacs should be determined using standard tritiated Dex as ligand, with competition using unlabelled Dex vs the protac.

Answer: This is another interesting suggestion. Our experiments of KH-103 coculture with DEX and Cort partially addresses the competition. They show that even in the absence of protein synthesis inhibition and despite potential competition against DEX KH-103 is sufficient to effectively deplete GR. This might be due to the event driven mode of action as opposed to the occupancy driven mode of action of DEX. Given these conceptually different modes of action, we are thus unsure whether such additional data would be a priority. Taking into consideration that it would require the assembly of an inactive PROTAC mimicking KH-103 structure, we thus deemed this experiment out of scope, while we tried to address most other comments.

The second part of the paper addresses specificity of action using RNA seq. The analysis appears sound, but the data visualisation in figs 7 and 8 makes it very hard for the reader to see anything, or draw any conclusions. Ideally MA plots are better than volcano plots, and the overlaps could be visualised better using Venn diagrams. I think the logical progression here is good, but the dip into public data relating to GR ChIP-SEQ tracks does not help much. I do wonder if there should be more effort to use the RNAseq to look for off target effects...the protac used is likely to target Ikaros proteins, and so can any such signature be looked for specifically?

Answer: We thank the reviewer for suggesting the addition of MA plots and Venn diagrams to the current set of graphs. We have now added those plots to the revised version and moved the Volcano plots to the supplement instead (Fig. 6b, c main text, Supplementary Fig. 10).

We totally agree that the addition of the ChIP seq data does not explain the clusters we depict – yet we find this negative finding also worthwhile mentioning. It clearly demonstrates that it is not the case that the occupancy driven inhibitors impact e.g., specific subsets of GR targets, based on presence or absence of response elements in regulatory regions.

The reviewer is right that the data additionally could be explored to look more for off-target effects, although we hope that the new proteomic data (along with the fact that KH-103 alone is having no effect on the transcriptome that is not also observed in the other blockers) offer more direct evidence of little side effects. Specifically, regarding the Ikaros proteins, since any effect on the proteins are bound to be post-transcriptional, we tried to find evidence for dysregulation of Ikaros targets. Of the best-known targets that are expressed in our system (CDK6, CDKN1A, FOXO1, LIG4, MYC below), many appear to be DEX-responsive, and as a consequence some are altered upon treatments, although less so using KH-103 than other blockers:

We further tried to investigate Ikaros targets in a more unbiased manner using motifs, however since the one that is (by far) most highly expressed in our system (IKZF5) does not have a characterised motif, we used the simple IKZF1 motif HYTCCCAV. Our analysis is thus based on the assumption (partially validated by known motifs) that all members of the family work via the same motif. We considered those as Ikaros target genes (expressed in our system), which show at least one well-supported (TSL1) protein-coding transcript with a motif instance -1kb/+200bp from TSS. While KH-103 does alter the expression of some targets (FKBP5, NBEAL2, TEF), this is not more than would be expected by chance (hypergeometric $p \sim 0.99$), most are also altered by other blockers (and by DEX), which also have larger overlap with the set (see Venn diagram below).

It is worthwhile mentioning that we do not expect a transcriptional off-target effect of KH-103 but are more concerned about post-transcriptional off-targets. A transcriptional off-target effect would be indistinguishable from a DEX/Cort effect, which we tested and do not observe in the KH-103 only condition. To address post-transcriptional off-target effects in a complementary approach though, we now also added a proteomics experiment that confirms the absence of any off-target protein depletion (Fig. 8c, line 500 main text, Supplementary Fig.20).

It is surprising that the protact has no effect on MR. Dex binds the MR, which they could demonstrate in their system, using over expressed MR, in HEK cells. Is it connected to the structural features of GR binding and differences in the LBD? Could they run some modelling to explore?

Answer: We agree with the reviewer that some effect on MR could have been plausible, since DEX can bind MR. The absence of effect can be explained by the differing dwelling time of DEX on MR versus DEX on GR (Reul et al., 2000). A shorter dwelling time of MR could be insufficient for ternary complex formation. To address this point in an adjacent way we a) performed the proteomic screen mentioned above and b) tried to perform modelling of the structures to infer some explanation of the absent effect on MR. Much to our disappointment MR was not detectable in the proteomic screen, and the modelling of MR was so far not successful either. We felt among all the suggested experiments, the overexpression of MR that does not reflect natural conditions, would be of less priority, but we are open, if the reviewer insists, to also perform this experiment.

The final fig is in two parts. The upper part may show a very tiny effect of Dex in terms of calcium transients. This is very unconvincing. The second part related to GR up regulation in pituitary. Again, this is an odd output. It is not clear why more typical Dex dependent changes in rodent physiology are not tested, eg anti-inflammation, or

effects on liver energy metabolic pathways. Its a shame to end the paper without a robust physiological end point.

Answer: We see that this reviewer would prefer other means of readouts for an application of KH-103. For calcium imaging we have substantiated our claims by providing further data and clarifications on the effects of DEX and KH-103, which we also explain in detail in our response to reviewer 1. We clarify that we also extracted a measure for the amplitude of peaks, which we however find less reliable than interval length, due to a non-linear add up of signal and saturation effects (please refer to our response to reviewer 1). Furthermore, we now provide representative traces and a raster plot depicting all measured peaks in every single cell over the duration of the experiment (Supplementary Fig. 21).

Amplitude

representative calcium traces

For a different output of rodent physiology, we are limited by our neuroscience focused animal experimentation licence, that does not allow us to induce inflammation/anti-inflammation. We hence turned towards the role of GR in the Hypothalamus Pituitary Adrenal (HPA) axis and the regulation of stress hormone levels. Following central infusion of KH-103 into the dorsal hippocampus via a cannula, we exposed animals to a restraint stress and measured systemic Cort levels during the rise and the recovery of the stress hormones. We observe an attenuated recovery of stress induced corticosterone levels that indicates a hippocampal GR mediated role in the termination of the stress response via negative feedback (Fig 9.f-h, line 550 main text, and below)

Additionally, we now provide further in vivo data on the direct KH-103 mediated GR regulation in pituitary in a dose dependent manner as suggested by reviewer 4 and a replication of the significant downregulation of GR in pituitary following intra peritoneal injection (Fig. 9c-e, line 542 main text and below).

Reviewer #4 (Remarks to the Author):

The work entitled " Harnessing PROTAC technology to combat stress hormone receptor activation " by Gazorpak et al. concerns the discovery of small-molecule-based inhibitors, novel catalytically-driven glucocorticoid receptor degrader. The study based on proteolysis-targeting chimera technology (PROTAC) which enables immediate and reversible depletion of glucocorticoid receptors. The idea is based on previous work with the estrogen receptor but is interesting due to the new target which is the GR receptor, therefore the work is in the area of searching for substances that block or modify the functions of the GR receptor original. The aim of the work is well defined. The activity of best compound KH-103 was compared to two currently available inhibitors. The effects of the compounds were measured in vitro in cell cultures and in the pituitary. KH-103 significantly inhibited, compared to existing inhibitors, gene and protein expression caused by GR receptor agonist.

The authors demonstrated the effect of KH-103 in vitro in 3 types of cell lines (HEC-293 cells, A549 and N2a mouse neuroblastoma cell line as well as in primary neuronal cells and also in the mouse pituitary. KH-103 showed robust GR degradation in vitro but a slight, although significant effect on the GR level in the pituitary after prolonged treatment. The compound produces a clear antagonistic effect on GR expression after 2 h following administration of DEX. The studies are of special interest for basic research into the functions of glucocorticoid and GR and the stress axis functions. However, the study is preliminary and more pharmacological data (eg. dose-response curve) and more replicates are needed. The effects of KH-103 on cells such as AtT20 or human corticotrophs would also be of interest.

Answer: We thank the reviewer for the positive evaluation of our work. Following the reviewers request we have enlarged our biological replicates for the in vivo application of KH-103 and added a dose response curve. Our dose response experiments clearly show a dose dependency on the depletion efficiency in vivo. These findings have been added to the revised manuscript (Fig.9c-d, line 542 main manuscript). We'd like to point out that the in vivo dosing requires a large amount of material and is thus not infinitely scalable as for a commercially available compound.

While we have conducted a broad range of additional experiments for this revision, we however felt that assessing the effectiveness of KH-103 in further cell lines would be of less priority. We are however open to reconsider, should the reviewer insist on its relevance.

Lenalidomide and dexamethasone is used in preclinical models and also use in clinical trials and treatment. How in this particular cell models a mixture of these substances works. It would be useful to know how a mixture of these substances performed in the experimental models studied. How does the effect of KH-103 differ from the compounds given together?

Answer: We thank the reviewer for suggesting this interesting and clinically relevant experiment. Following the reviewer's recommendation, we exposed cells to either KH-103, DEX, Lenalidomide or different combinations. Our data clearly demonstrate no detectable impact on the performance of KH-103 to deplete GR by the presence of Lenalidomide alone or in combination with DEX. These data have been added to the revised manuscript (Fig.4c, line 284 main manuscript).

RNAseq studies were performed in A549 cell cultures upon exposure of DEX treatment. The material for RNAseq was acquired at 3 different time points (2,12 and 18 h). Positive correlations of results of Log FC obtained in two laboratories are presented. The correlations from two different labs don't seem to be needed.

Answer: We are sorry if our labels caused confusion - in fact the comparison refers to our data and data from the Reddy lab. This has now been clarified in the revised versions figure caption (lines 1437 & 1439 main manuscript) and supplemented with Venn diagrams as described below.

Perhaps a Venn diagram would be more informative than a correlation study?

Answer: We have added a Venn diagram to the revised manuscript (Supplementary Fig 8, 11b, 13, 18b). We note, however, that Venn diagrams tend to be misleading for the comparison of differentially-expressed genes, giving a false impression of specificity that is often fragile to the threshold used. In our opinion, the comparison of fold changes offers a considerably better picture of the reproducibility of experiments.

Figure 6. Fig. 6 d show the logFC correlations after 2 and Fig. 6e the correlation between the results obtained after 12 and 18 h of DEX exposure. Fig. 6b is described as a correlation and shows the Volcano plot. Fig 6 f-i is missing from this figure. This figure should be corrected.

Answer: We appreciate that the reviewer spotted this mistake- we have fixed Fig 6 and its legend.

Figure 3 shows the KH-103-mediated nuclear translocation of GR. The studies were carried out on HEC293 cells and studies on the regulation of expression of two selected genes dependent on the GR receptor on a cell line are being added. Why not in Hec293?

Answer: The reviewer is right that we could have done the gene regulation studies also in HEK293 cells. We show in Fig 3B that translocation upon KH-103 also happens in A549 cells, which is the cell line we chose to do all transcriptome wide experiments on, due to its already well established strong response to DEX and the availability of reference datasets in the very same cell line (McDowell et al., 2018). We do detect the same transcriptional isoforms in HEK293 and A549 cells, including alpha, gamma and P and based on the literature (Morgan

et al., 2016; Pujols et al., 2012) also expect GR-beta to be present (see also Figure above together with response to reviewer number 2 and newly added Supplementary Fig. 5). We thus do not expect KH-103's performance and potential effects on gene expression regulation to be overly distinct in HEK293 cells.

The RNA-Seq analysis was done with the right tools (salmon + SVA + edgeR) and looks reasonable. They use the phrase FDR in the Figs, which suggests that Multiple-Testing correction has been made.

Answer: We apologise if our description was not clear enough. We indeed applied multiple testing corrections and have now clarified this in the figure legends of the revised version (Figure legends 6,7,8).

Since edgeR was used, it should be stated which type of FDR correction was applied. It is difficult to find the size of the groups (that is, how many replicates and which ones were used).

Answer: We used the default FDR correction (Benjamini Hochberg) and had 4 biological replicates in each group (besides in the 18 hours DMSO group, that had 3 replicates only - we did not exclude any replicates. Since untreated and DMSO conditions did not show changes, these were combined into one group. In our heatmaps each column represents one replicate. We have now added this information to the revised version (Figure legends, Figures 6a, Supplementary Fig. 7).

Clear and a correct diagram explaining the "study design" for the RNA-Seq should be presented.

Answer: We thank the reviewer for this suggestion and have added said study design diagram (supplementary Fig. 7 and attached below)

Perhaps more information on this subject can be found at <https://github.com/ETHZ-INS/Glucocorticoid-protacs> but the page is not available.

Answer: We failed in providing a reviewer token - the information is now freely available.

The filtering genes in the differential analysis was done only by removing those genes with the number of reads less than 20: "filtering genes using filterByExpr with a minimum count of 20" (the question is why 20?). However, there is nothing about filtering by logFC. The pictures also show that this was not done. The recommendation for microarrays but repeated for sequencing is that a limited trust is applied to the data when $|\logFC| < 1$ (Su et al., 2014).

Answer: The reviewer is presumably drawing this conclusion from Figure 6, where we compared the fold changes of all genes passing FDR, but in fact for all other analyses a logFC threshold was employed. We agree that such thresholding is advisable and apologise if this was not sufficiently described. Given the low variability of our system (the SEQC samples are considerably more variable), and our interest in catching potential off-target effects, we employed a threshold of 20% change (i.e. $|\logFC| > \log_2(1.2)$). This is admittedly lower than the very stringent $|\logFC| > 1$ suggested by the reviewer; therefore, to ensure the robustness of our results we repeated the analyses using the more stringent threshold (analysis available in the repositories, see especially the "DEA_stringent" file of MG_A649 repository). While the number of filtered DEGs is obviously smaller, our main observations are confirmed (Figures A,B below). Specifically, we again observe a slightly better inhibition by KH-103 when applied prior to DEX (panel A), with a less efficient reversal (panel B).

We further confirmed that treatment with KH-103 alone triggers fewer side-effects than either MIF or CORT113176. This information is now also available in the supplement under Fig. 16.

Finally, we also observe using the stringent logFC threshold that GR binding at the promoter explains only a minority of the DEX-responsive genes (see Figure below).

Furthermore, the "vulcano-plot" was used, which is misleading for high-throughput technologies where all genes are studied. P-value/q-value/FDR should only be used to cut off the statistically insignificant, but further "quantification" or other assessment should be based on a comparison of logFC and average expression. Therefore, MA-plot rather than Vulcano-plot is recommended.

Answer: We followed the reviewers suggestion and substituted the volcano plots with MA plots in addition to the volcano- plots (Fig. 6b,c), the volcano plots were moved to the supplement Fig. 10a,b.

Some semantic issue. On line 352 they write: "Analyzing differentially expressed genes (DEGs) revealed thousands of significantly affected transcripts". In principle RNA-Seq measure the expression of each of the alternative gene transcripts, but here the analysis is performed at the gene level. Although Salmon gives expression levels for alternative transcripts as a result, they themselves write that "Counts were aggregated to gene-level using the tximport package". Therefore, the sentence on line 352 is confusing.

Answer: We absolutely agree with the reviewer and have amended the phrasing accordingly (line 481 main text).

The broken code page and the lack of information about the group size, i.e. what exactly the RNA-Seq experiment looked like, are critical things to complete.

It would also be required to check whether if the authors apply the $|\log FC| < 1$ filter, whether their results will not change.

Answer: As mentioned above we have added this information to the revised manuscript, have amended the code page and clarified that a more stringent filter criterion as suggested had already been implemented before, while we here also provided the results using the recommended filter. We thank the reviewer for pointing out these important points that slipped our attention and that certainly contribute substantially to the improvement of the revised version of the manuscript.

References

- Chinenov, Y., Coppo, M., Gupte, R., Sacta, M. A., & Rogatsky, I. (2014). Glucocorticoid receptor coordinates transcription factor-dominated regulatory network in macrophages. *BMC Genomics*, *15*(1), 656. <https://doi.org/10.1186/1471-2164-15-656>
- Dana, H., Sun, Y., Mohar, B., Hulse, B. K., Kerlin, A. M., Hasseman, J. P., Tsegaye, G., Tsang, A., Wong, A., Patel, R., Macklin, J. J., Chen, Y., Konnerth, A., Jayaraman, V., Looger, L. L., Schreier, E. R., Svoboda, K., & Kim, D. S. (2019). High-performance calcium sensors for imaging activity in neuronal populations and microcompartments. *Nature Methods* *2019* *16*:7, *16*(7), 649–657. <https://doi.org/10.1038/s41592-019-0435-6>
- DeFranco, D. B., Qi, M., Borrer, K. C., Garabedian, M. J., & Brautigan, D. L. (1991). Protein Phosphatase Types 1 and/or 2A Regulate Nucleocytoplasmic Shuttling of Glucocorticoid Receptors. *Molecular Endocrinology*, *5*(9), 1215–1228. <https://doi.org/10.1210/MEND-5-9-1215>
- Estévez-Priego, E., Moreno-Fina, M., Monni, E., Kokaia, Z., Soriano, J., & Tornero, D. (2023). Long-term calcium imaging reveals functional development in hiPSC-derived cultures comparable to human but not rat primary cultures. *Stem Cell Reports*, *18*(1), 205–219. <https://doi.org/10.1016/J.STEMCR.2022.11.014>
- Faria-Pereira, A., Temido-Ferreira, M., & Morais, V. A. (2022). BrainPhys Neuronal Media Support Physiological Function of Mitochondria in Mouse Primary Neuronal Cultures. *Frontiers in Molecular Neuroscience*, *15*. <https://doi.org/10.3389/FNMOL.2022.837448/FULL>
- Gadd, M. S., Testa, A., Lucas, X., Chan, K. H., Chen, W., Lamont, D. J., Zengerle, M., & Ciulli, A. (2017). Structural basis of PROTAC cooperative recognition for selective protein degradation. *Nature Chemical Biology* *2017* *13*:5, *13*(5), 514–521. <https://doi.org/10.1038/nchembio.2329>

- Galigniana, M. D., Housley, P. R., DeFranco, D. B., & Pratt, W. B. (1999). Inhibition of Glucocorticoid Receptor Nucleocytoplasmic Shuttling by Okadaic Acid Requires Intact Cytoskeleton. *Journal of Biological Chemistry*, 274(23), 16222–16227. <https://doi.org/10.1074/JBC.274.23.16222>
- Geng, J., Tang, Y., Yu, Z., Gao, Y., Li, W., Lu, Y., Wang, B., Zhou, H., Li, P., Liu, N., Wang, P., Fan, Y., Yang, Y., Guo, Z. V., & Liu, X. (2022). Chronic Ca²⁺ imaging of cortical neurons with long-term expression of GCaMP-X. *ELife*, 11. <https://doi.org/10.7554/ELIFE.76691>
- Madan, A. P., & DeFranco, D. B. (1993). Bidirectional transport of glucocorticoid receptors across the nuclear envelope. *Proceedings of the National Academy of Sciences*, 90(8), 3588–3592. <https://doi.org/10.1073/PNAS.90.8.3588>
- McDowell, I. C., Barrera, A., D'Ippolito, A. M., Vockley, C. M., Hong, L. K., Leichter, S. M., Bartelt, L. C., Majoros, W. H., Song, L., Safi, A., Koçak, D. D., Gersbach, C. A., Hartemink, A. J., Crawford, G. E., Engelhardt, B. E., & Reddy, T. E. (2018). Glucocorticoid receptor recruits to enhancers and drives activation by motif-directed binding. *Genome Research*, 28(9), 1272–1284. <https://doi.org/10.1101/GR.233346.117/-/DC1>
- Morgan, D. J., Poolman, T. M., Williamson, A. J. K., Wang, Z., Clark, N. R., Ma'ayan, A., Whetton, A. D., Brass, A., Matthews, L. C., & Ray, D. W. (2016). Glucocorticoid receptor isoforms direct distinct mitochondrial programs to regulate ATP production. *Scientific Reports*, 6. <https://doi.org/10.1038/SREP26419>
- Pujols, L., Mullol, J., Pérez, M., Roca-Ferrer, J., Juan, M., Xaubet, A., Cidlowski, J. A., & Picado, C. (2012). Expression of the Human Glucocorticoid Receptor α and β Isoforms in Human Respiratory Epithelial Cells and Their Regulation by Dexamethasone. *https://doi.org/10.1165/Ajrcmb.24.1.4024*, 24(1), 49–57. <https://doi.org/10.1165/AJRCMB.24.1.4024>
- Reul, J. M. H. M., Gesing, A., Droste, S., Stec, I. S. M., Weber, A., Bachmann, C., Bilang-Bleuel, A., Holsboer, F., & Linthorst, A. C. E. (2000). The brain mineralocorticoid receptor: greedy for ligand, mysterious in function. *European Journal of Pharmacology*, 405(1–3), 235–249. [https://doi.org/10.1016/S0014-2999\(00\)00677-4](https://doi.org/10.1016/S0014-2999(00)00677-4)
- Su, Z., Łabaj, P. P., Li, S., Thierry-Mieg, J., Thierry-Mieg, D., Shi, W., Wang, C., Schroth, G. P., Setterquist, R. A., Thompson, J. F., Jones, W. D., Xiao, W., Xu, W., Jensen, R. V., Kelly, R., Xu, J., Conesa, A., Furlanello, C., Gao, H., ... Shi, L. (2014). A comprehensive assessment of RNA-seq accuracy, reproducibility and information content by the Sequencing Quality Control Consortium. *Nature Biotechnology* 2014 32:9, 32(9), 903–914. <https://doi.org/10.1038/nbt.2957>
- Yada, Y., Mita, T., Sanada, A., Yano, R., Kanzaki, R., Bakkum, D. J., Hierlemann, A., & Takahashi, H. (2017). Development of neural population activity toward self-organized criticality. *Neuroscience*, 343, 55–65. <https://doi.org/10.1016/J.NEUROSCIENCE.2016.11.031>
- Zhu, H., Li, J., Li, Y., Zheng, Z., Guan, H., Wang, H., Tao, K., Liu, J., Wang, Y., Zhang, W., Li, C., Li, J., Jia, L., Bai, W., & Hu, D. (2021). Glucocorticoid counteracts cellular mechanoresponses by LINC01569-dependent glucocorticoid receptor-mediated mRNA decay. *Science Advances*, 7(9), 9923–9947. https://doi.org/10.1126/SCIADV.ABD9923/SUPPL_FILE/ABD9923_TABLE_S1.XLSX

Reviewers' Comments:

Reviewer #1:

Remarks to the Author:

The authors address all my comments

Reviewer #2:

Remarks to the Author:

The authors made a good effort to revise the manuscript.
I do not have further comments.

Reviewer #3:

Remarks to the Author:

The paper is improved, and they have done a great job addressing referee questions.

Reviewer #4:

Remarks to the Author:

The paper under review was substantially improved, and the authors responded to most of the suggestions and comments included in the previous review.

Fig. 9 has been changed Fig. 9c has been withdrawn and replaced by new Fig 9c showing the dose-dependent effect of KH103 on GR levels in the pituitary gland of mice. However, the data presented is preliminary. The results were obtained on a small number of mice (n=3) of each dose of KH103 and for a dose of 10 mg/kg (n= 2). The schematic depiction of administration of KH103 to mice is not fully understood. How are the studies conducted on two groups of mice?

One group received KH103 intraperitoneally, and the other subcutaneously? The Fig also contains obvious errors. On the Y-axis should be GR level but not a percentage of the vehicle and the description on the X axis should also be corrected. Why is the route of administration showed at the top of the figure?

Fig. 9e shows the effect of 44mg/kg KH=103 i.p. (why not for example 50 mg/kg?) on GR levels in the pituitary upon intraperitoneal injection. A single intraperitoneal dose significantly inhibited GR levels (a decrease of approximately 20%) when the more complicated s.c. led to a significant decrease only at around 60%. These pharmacological studies are far from the quality of in vitro studies and, because of the importance of such studies, it is necessary to organize them properly.

The effect of KH-103 on the level of GR in the pituitary is shown in relative terms as a percentage of the effect of the vehicle and not in absolute numbers. It would be necessary to show the original immunoblots as shown in an in vitro study. The fragment in the corrected manuscript indicates that such data is presented in the supplementary file, but it does not provide them with the revision.

Reviewer #1 (Remarks to the Author):

The authors address all my comments

Reviewer #2 (Remarks to the Author):

The authors made a good effort to revise the manuscript.
I do not have further comments.

Reviewer #3 (Remarks to the Author):

The paper is improved, and they have done a great job addressing referee questions.

Answer: We are very pleased that reviewer 1-3 found our resubmitted manuscript adequate for publication.

Reviewer #4 (Remarks to the Author):

The paper under review was substantially improved, and the authors responded to most of the suggestions and comments included in the previous review.

Answer: We thank the reviewer for acknowledging the amount of effort we put into addressing most of the concerns raised by the reviewers.

Fig. 9 has been changed Fig. 9c has been withdrawn and replaced by new Fig 9c showing the dose-dependent effect of KH103 on GR levels in the pituitary gland of mice. However, the data presented is preliminary. The results were obtained on a small number of mice (n=3) of each dose of KH103 and for a dose of 10 mg/kg (n= 2). The schematic depiction of administration of KH103 to mice is not fully understood. How are the studies conducted on two groups of mice?

One group received KH103 intraperitoneally, and the other subcutaneously?

Answer: We apologize very much that the schematic description of some of the in vivo experiments (previous Figure 9c) confused the reviewer. We understand the ambiguous interpretation possibilities. Therefore, we have split the schematic in 2 to explain that one group of mice received solely subcutaneous injections, while in a separate experiment we administered the drug intraperitoneally. In both experiments mice were treated on 2 consecutive days and pituitary tissue was collected 4 hours following the second injection. Following a request in the first round of revision, we could herewith reproduce the depletion efficiency of GR by KH-103 at 44mg/kg using intraperitoneal injection, in an additional, independent set of animals. We therefore argue that there this is conclusive evidence for this dosage to be effective when administered i.p..

Additionally, we provided the requested dose-response curve. For this we chose the subcutaneous administration route, a suggestion by Dr. Behnam Nabet, who had observed increased efficiency of depletion with in vivo administration of another PROTAC via this route. The dose response experiment overall uses a substantial amount of animals (and thus compound) across different dosages, that follow the expected dose dependent depletion effect of GR and therefore should also not be considered preliminary.

The Fig also contains obvious errors. On the Y-axis should be GR level but not a percentage of the vehicle and the description on the X axis should also be corrected. Why is the route of administration showed at the top of the figure?

Answer: We thank the reviewer for spotting the inconsistency in the way we depict the label of the Y-axis. We have now amended the Y-axis and the route of administration is no longer shown at the top of the figure.

Fig. 9e shows the effect of 44mg/kg KH=103 i.p. (why not for example 50 mg/kg?) on GR levels in the pituitary upon intraperitoneal injection. A single intraperitoneal dose significantly inhibited GR levels (a decrease of approximately 20%) when the more complicated s.c. led to a significant decrease only at around 60%. These pharmacological studies are far from the quality of in vitro studies and, because of the importance of such studies, it is necessary to organize them properly.

Answer: The dosage chosen in the i.p. experiment was based on the dosage used in the first submitted manuscript, with the aim to reproduce the depletion of GR by KH-103 at this dosage. Initially, this dosage had purely practical reasons and was based on the maximally achieved solubility with the stock solution we had at hand. For the follow up dose-response curve we had a new batch of KH-103 PROTAC at our disposition and therefore decided to use more intuitive increments of dosages. Please note that unlike for a commercially available drug the amounts required for in vivo experiments are not as readily available, since we have to synthesize the drug ourselves. The effects using the subcutaneous route are stronger than the effects achieved at similar dosage following i.p. injections in line with the prediction of our advisor Dr. Behnam Nabet and with known differences in absorption between different administration routes and their interaction with the elimination of the drug. For many drugs the highest concentration in circulation (Cmax) reached is higher following i.p. than s.c. injections yet at the same time the time it takes to reach this concentration is slower following i.p. than following s.c. (Shoyaib et al 2019 [10.1007/s11095-019-2745-x](https://doi.org/10.1007/s11095-019-2745-x)). Hence the optimal timepoint to observe an effect following administration likely differs between intraperitoneal and subcutaneous injection. Additionally, GR has genomic and non-genomic autoregulatory action, that also modulates GR levels potentially triggering compensatory stimulation of GR production in the pituitary (Gupta et al. 2007, <https://tbiomed.biomedcentral.com/articles/10.1186/1742-4682-4-8>). Considering these complex dynamics higher depletion of GR might be expectable at a shorter time frame following intraperitoneal administration. We now also discuss this important considerations in the revised version of the manuscript(Lines: 565-568 &616-628)

The effect of KH-103 on the level of GR in the pituitary is shown in relative terms as a percentage of the effect of the vehicle and not in absolute numbers. It would be necessary to show the original immunoblots as shown in an in vitro study. The fragment in the corrected manuscript indicates that such data is presented in the supplementary file, but it does not provide them with the revision.

Answer: We agree with the reviewer that showing the original immunoblots is an elegant way to support the data depicted in the bar graphs. For space reasons the blots were previously contained in the supplementary Figures, yet now upon overall rearrangement of Figure 9 we have included them in the main figure “d” & “f”.